# Evidence of conditioned behavior in amoebae

Ildefonso M. De la Fuente[1,2], Carlos Bringas[3], Iker Malaina[2], María Fedetz[4], Jose Carrasco-Pujante[3], Miguel Morales[5], Shira Knafo[5,6,7], Luis Martínez[2,8], Alberto Pérez-Samartín[9], José I. López[10], Gorka Pérez-Yarza [3] & María Dolores Boyano [3]

Associative memory is the main type of learning by which complex organisms endowed with evolved nervous systems respond efficiently to certain environmental stimuli. It has been found in different multicellular species, from cephalopods to humans, but never in individual cells. Here we describe a motility pattern consistent with associative conditioned behavior in the microorganism *Amoeba proteus*. We use a controlled direct-current electric field as the conditioned stimulus, and a specific chemotactic peptide as the unconditioned stimulus. The amoebae are capable of linking two independent past events, generating persistent loco-motion movements that can prevail for 44 min on average. We confirm a similar behavior in a related species, *Metamoeba leningradensis*. Thus, our results indicate that unicellular organisms can modify their behavior during migration by associative conditioning.

[1] Department of Nutrition, CEBAS-CSIC Institute, Espinardo University Campus, Murcia 30100, Spain. [2] Department of Mathematics, Faculty of Science and Technology, University of the Basque Country, UPV/EHU, Leioa 48940, Spain. [3] Department of Cell Biology and Histology, Faculty of Medicine and Nursing, University of the Basque Country, UPV/EHU, Leioa 48940, Spain. [4] Department of Cell Biology and Immunology, Institute of Parasitology and Biomedicine "López-Neyra", CSIC, Granada 18016, Spain. [5] Biophysics Institute, CSIC-UPV/EHU, University of the Basque Country (UPV/EHU), Leioa 48940, Spain. [6] Ikerbasque, Basque Foundation for Science, Bilbao 48013, Spain. [7] Department of Physiology and Cell Biology, Faculty of Health Sciences, and The National Institute for Biotechnology in the Negev, Ben-Gurion University of the Negev, Beer-Sheva, Israel. [8] Basque Center of Applied Mathematics (BCAM), Bilbao 48009, Spain. [9] Department of Neurosciences, Faculty of Medicine and Nursing, University of the Basque Country, UPV/EHU, Leioa 48940, Spain. [10] Department of Pathology, Cruces University Hospital, Biocruces-Bizkaia Health Research Institute, University of the Basque Country (UPV/EHU), Barakaldo 48903, Spain. Correspondence and requests for materials should be addressed to I.F. (email: mtpmadei@ehu.eus)

One of the most remarkable accomplishments in the field of neuroscience is the description of essential principles that define the basic forms of associative memory. This fundamental cognitive property occurs through complex biological mechanisms by which the connection between two previously unrelated stimuli, or a behavior and a stimulus, is learned; when such process takes place, it is assumed that the association of these stimuli is stored in a memory system[1]. For centuries, different thinkers have shaped a very plentiful and venerable history of research on basic learning processes. The combined work of philosophers, naturalists, physiologists, and life scientists has set the baseline upon which the modern learning theory currently stands[2]. The most basic type of associative learning is the classical conditioning developed by the Nobel Prize Laureate Ivan Pavlov, who established the first systematic study of the fundamental principles of associative memory. In his studies, after an appropriate conditioning, dogs deprived of food were able to exhibit a consequent response -salivation- when a bell rung[3].

Associative conditioning is ubiquitous in complex organisms endowed with evolved nervous systems, including all major vertebrate taxa and several invertebrate species[4]. This complex process can also be reproduced and analyzed in artificial neural networks and different computational models[5]. Conditioned learning confers to the organisms the ability to adapt to everchanging environments and is considered a milestone for life's survival. Despite its importance, associative conditioning has never been observed in individual cells.

In order to determine whether associative conditioned responses are involved in systemic cellular behaviors, we analyzed the movement trajectories of *Amoeba proteus* under two external stimuli by using an appropriate electric field as the conditioned stimulus and a specific peptide as chemo-attractant.

*Amoebae* represent an immensely diverse family of eukaryotic cells that can be found in nearly all habitats and constitute the major part of all eukaryote lineages[6]. Concretely, *Amoeba proteus* is a large free-living predatory amoeba with a notable capacity to detect and respond to chemical and physical cues allowing it to locate and consume near prey organisms such as bacteria and other protists.

These cells are able to migrate on flat surfaces and in three-dimensional substrate by a process known as amoeboid movement, which consists in pseudopodia extensions, cytoplasmic streaming, and flowing into these extensions changing permanently the cellular shape[7].

Amoeboid locomotion represents one of the most widespread forms of cell motility and constitutes the typical way of locomotion in broad range of adherent and suspended eukaryotic cell types[7]. In mammalian cells, amoeboid locomotion is vital for multiple physiological processes as the development of the embryo[8], the action of the immune system[9] and the repair of wounds[10]. Likewise, it is also responsible for the spread of malignant tumors[11].

The large free-living amoeba, *Amoeba proteus*, has served as a classic unicellular organism in many investigations for more than one hundred years[12,13], mainly as a cellular model to study cell motility, membrane and cytoskeleton function, and the role of the nucleus[14–16]. However, despite the many investigations carried out so far, numerous biological aspects of this organism still remain poorly studied. On the other hand, diverse experimental studies have shown that *Amoeba proteus* exhibit robust galvanotaxis[17], a directed movement in response to an electric field; in fact, it has been described that practically 100% of the amoebae migrate towards the cathode for long periods of time under a strong direct-current electric field in a range between 300 mV/mm and 600 mV/mm. Likewise, amoebae are known to display

chemotactic behaviors; in particular, the peptide nFMLP, typically secreted by bacteria, is able to provoke a strong chemotactic response in many different types of cells. The presence of this peptide in the environment may indicate to the amoeba that food organisms might be nearby[18]. Given the large number of investigations carried out on this organism, the robustness in their behavior, easy handling in the laboratory, the relatively fast rate of migration (cells move at ~300 μm/min[19]) and the well documented sensitivity to electric fields and chemoattractants, we have chosen *Amoeba proteus* as the experimental study species in our work.

Here, we describe the emergence of an associative conditioned behavior in *Amoeba proteus* which corresponds to a new type of systemic migration pattern in the cell. This conditioned migration behavior seems to be an evidence of a primitive type of associative memory in a unicellular organism. In a preliminary study, we also confirm a similar behavior in a related species, *Metamoeba leningradensis*.

## Results

**Experimental setup**. All our experiments have been carried out on a specific set-up that allowed us to expose the amoebae to both stimuli -galvanotaxis and chemotaxis- simultaneously. This system consists of two standard electrophoresis blocks, about 17.5 cm long, one directly plugged into a normal power supply and a second one connected to the first one via two agar bridges that transfer the current from one block to the other while preventing the direct contact of both the anode and the cathode with the medium where the cells were located (see Fig. 1, Supplementary Data 1, and data available in Methods section). On the central platform of the second electrophoresis block, we placed the experimental chamber, a sliding glass structure that enabled the creation of a laminar flux which not only allowed the electric current to pass through, but also generated an nFMLP peptide gradient that the amoebae were able to detect and respond to. In addition, when the sliding glass structure was opened, the placement and collecting of the cells was possible. We confirmed the establishment of the nFMLP gradient by the direct measurement of fluorescein-tagged peptide concentration with a plate reader. As shown in Fig. 2, the concentration of peptide in the middle part of the glass chamber (where the amoebae are placed) increases immediately following the flow establishment (within 2 min the concentration rises from zero to approximately 0.2 μM) and this concentration increases further (to 0.6 μM) for at least 30 min.

In the experiments, the cells were placed in the middle of the glass set-up and their displacements were monitored in small groups (see Methods section), being the individual trajectories recorded during periods of 30 min by using a camera connected to a microscope. The migration of 615 *A. proteus* and 210 *M. leningradensis* was quantitatively analyzed (Supplementary Data 2). All the experiments were carried out in Chalkley's medium, a standard, nutrient-free saline medium at ambient temperature.

**Cellular migration of *A. proteus* in the absence of stimuli**. First, we recorded the locomotion trajectories of 50 amoebae (experimental replicates: 7, number of cells per replicate: 5–11) without any external influences (Fig. 3a). Under this condition, cell migration can be described as a correlated random motion characterized by low intrinsic directionality and progressive decreasing over time of the initial direction of migration[20]. In Fig. 3a, a representative example of amoebae locomotion in the absence of stimuli is depicted; cells exhibited significant changes in the movement guidance, and after 30 min they had explored practically all the directions of the experimentation chamber. The

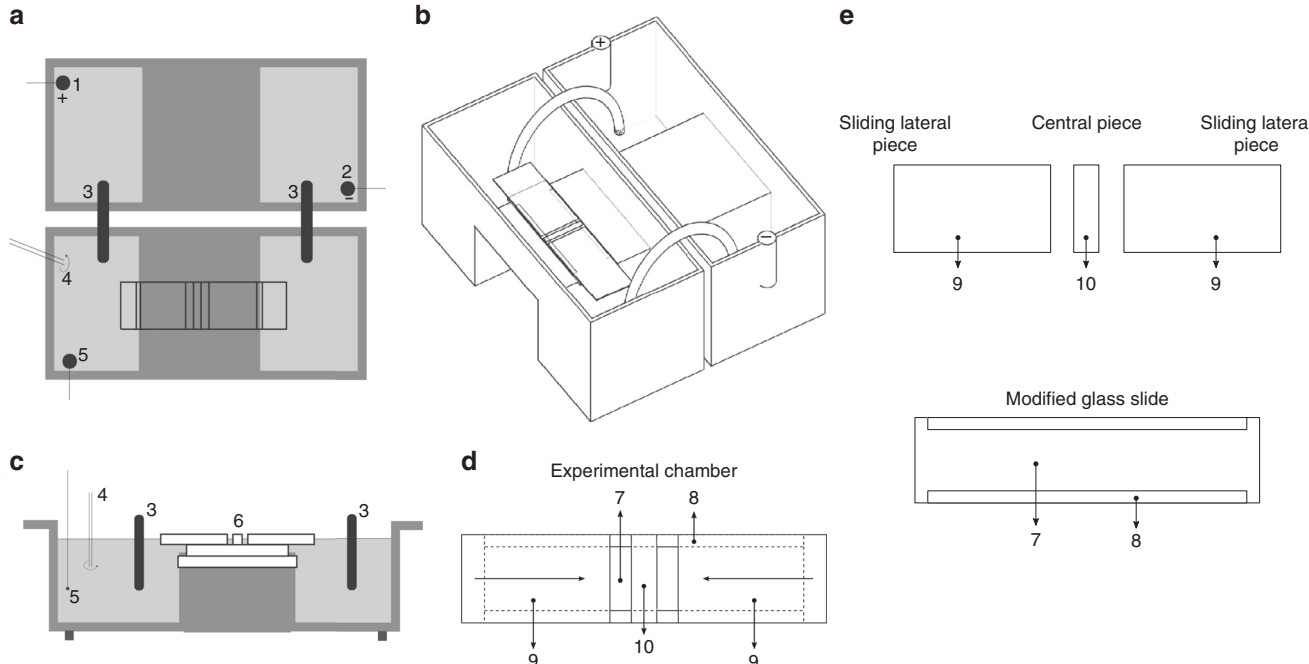

**Fig. 1** Experimental set-up. **a**, **c** illustrate the top and lateral views of the experimental system (two standard electrophoresis blocks). 1: anode; 2: cathode; 3: agar bridges, 10–12 cm long; 4: chemotactic peptide; 5: probe electrode used to monitor the electric field; 6: glass structure (experimental chamber). **b** contains an isometric view of the experimental set-up (data available in Methods section). **d** corresponds to the top view of the glass structure in which the cells are placed. 7: standard slice glass 75 × 25 mm; 8: longitudinal strip of glued cover glasses 0.1 mm tall; 9: sliding lateral pieces of cover glass, each 4 cm long; 10: central piece of cover glass, about 3 mm wide under which the cells are placed. The experimental chamber consists in a sliding glass structure **d** in which the sliding pieces **d**, **e** can be displaced in the longitudinal direction. This way, when the sliding pieces are closed an inner laminar flux is available in the chamber and, when they are opened, the placement and collecting of the cells is possible

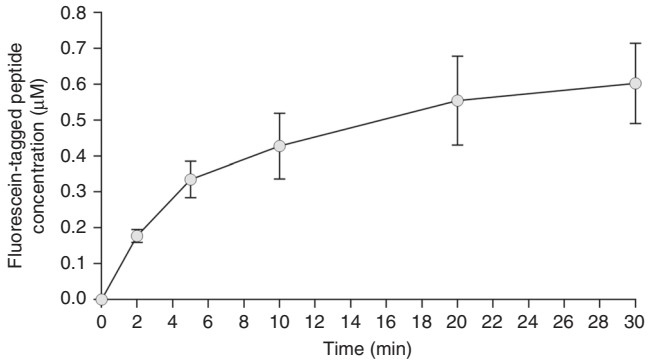

**Fig. 2** Fluorescein-tagged peptide concentration in the middle part of the laminar chamber flux as determined with a plate reader. The data represents the Mean ± SEM of 6 measurements (taken at 0, 2, 5, 10, 20, and 30 min) from 3 separate experiments

directionality of each cell was quantified by the cosine of the displacement angle[17] (see Methods section), and the quantitative results showed that the values ranged between −0.987 and 1, being −0.125/1.55 the media/interquartile range (IQ), which indicated that in the absence of stimulus, these cells moved randomly without any defined guidance. In addition, the analysis of the distribution of displacement angles (i.e., the angle formed between the origin and the end of the movement, measured in radians) also confirmed no preference towards a certain direction (Fig. 3d).

**Cell behavior of *Amoeba proteus* in an electric field**. Next, the galvanotactic locomotion of 50 cellular trajectories (experimental replicates: 8, number of cells per replicate: 4–8) was analyzed under an external, controlled direct-current electric field of about 300 mV/mm (Methods section). Our experimental observations (Fig. 3b) indicated that practically all the amoebae migrated towards the cathode for 30 min. These results matched with other previously reported experiments[17]. Cellular locomotion under this electric condition was characterized by stochastic movements with robust directionality, and cells exhibited a locomotion pattern tending to move in the direction of the immediately preceding movement by conserving their polarity in time towards the cathode. Taking as reference the experiment of Fig. 3b, the quantitative analysis indicated that the values of the cosines of displacements were distributed between 0.037 and 0.999 (0.993/0.03 median/IQ) (Fig. 3e). This result verified that a unique fundamental behavior characterized by an unequivocal directionality towards the cathode had emerged in the experimental system. The significance of our analysis was validated with a non-parametric test (Wilcoxon rank-sum test) comparing the distributions of the cosines of the displacement in both situations, without stimulus and under the presence of the electric field. The test ($p = 10^{-14}$; $Z = 7.442$, Wilcoxon rank-sum test) corroborated that the behaviors without and with the stimulus (the electric field) were significantly different.

**Directionality under chemotactic gradient (chemotaxis)**. Here, the behavior of 50 amoebae (experimental replicates: 10, number of cells per replicate: 4–6) was analyzed during biochemical guidance by exposing the cells for 30 min to an nFMLP peptide gradient placed in the left side of the set-up. The experiment showed that 86% of exposed cells migrated towards the chemotactic gradient (Fig. 3c). In other words, the chemical gradient in the environment provoked in most amoebae a systemic behavior characterized by stochastic locomotion movements with robust

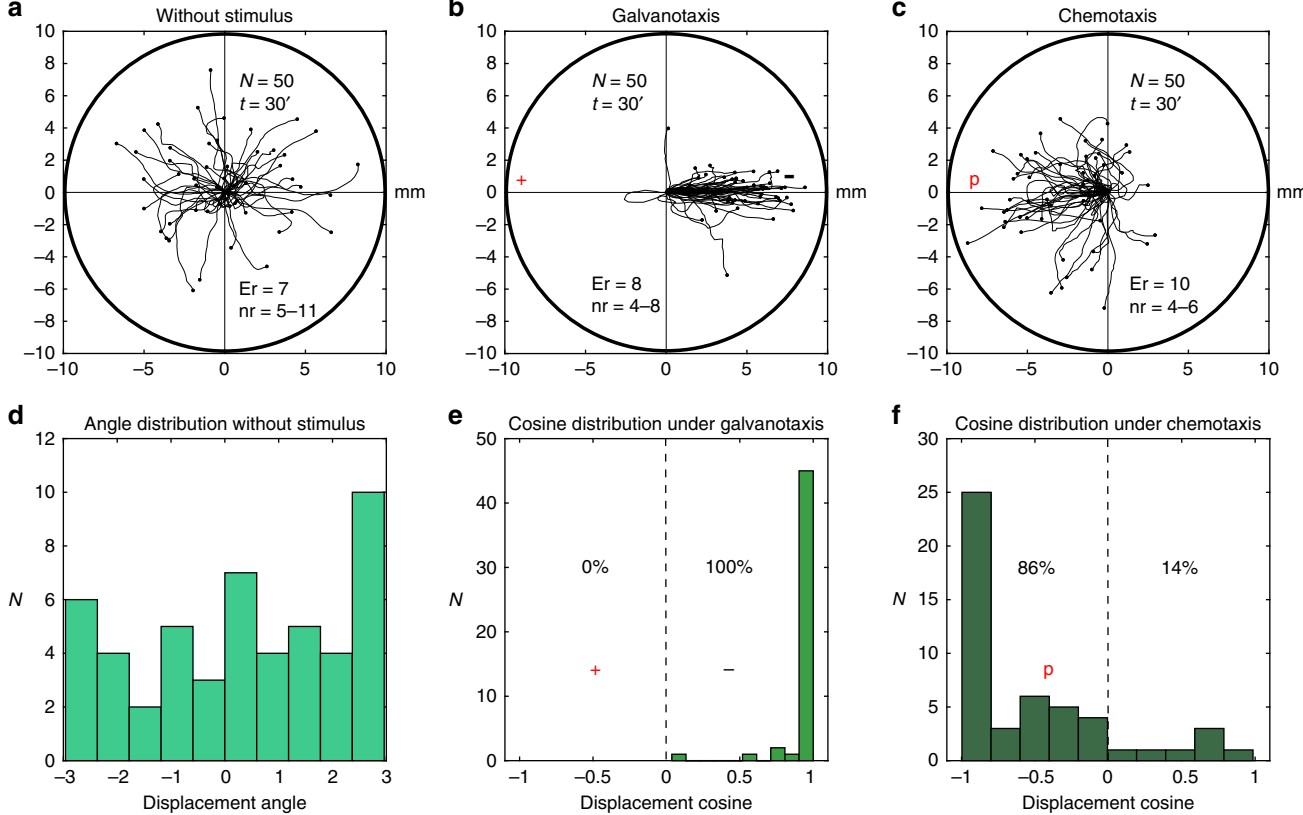

**Fig. 3** Migration trajectories of *Amoebae proteus* under three basic and independent experimental conditions: without stimulus, galvanotaxis and chemotaxis. **a** Trace – without stimulus. Without stimulus the cells practically explored all the directions of the experimentation chamber. **b** Trace – galvanotaxis. Under galvanotaxis conditions practically all the amoebae migrated towards the cathode. **c** Trace – chemotaxis. Under chemotaxis conditions, 86% of the cells migrated towards the chemotactic gradient. **d** Plot of displacement angle for a. Distribution of displacement angles (i.e., the angle formed between the origin and the end of the movement, measured in radians) for the trajectories without stimulus (**a**). No preference towards a certain direction was appreciated. **e** Plot of displacement angle for b. Distribution of the cosines of displacement angles for the trajectories under galvanotaxis (**b**). 100% of the displacement cosines were bigger than 0, indicating a strong directionality towards the cathode. **f** Plot of the cosines of displacement angles for trajectories under chemotaxis (**c**). "*N*" total number of cells, "Er" experimental replications, "nr" number of cells per replication, "*t*" time of galvanotaxis or chemotaxis, "p" chemotactic peptide (nFMLP), "+" anode, "−" cathode. Both the *x* and *y* axis show the distance in mm, and the initial location of each cell has been placed at the center of the diagram

directionality towards the attractant stimulus (the peptide)[21]. The cosines of the displacement angles of individual trajectories ranged between −0.997 and 0.986 (−0.825/0.72 median/IQ) (Fig. 3f). This result indicated that a single fundamental behavior characterized by a movement towards the peptide prevailed in the cells. The comparison between the cosine values obtained with and without chemotactic stimulus ($p = 10^{-4}$; $Z = −3.878$, Wilcoxon rank-sum test) on one hand, and between the cosine values with chemotactic gradient and with the presence of electric field ($p = 10^{-17}$; $Z = 8.428$, Wilcoxon rank-sum test) on the other, corroborated that the systemic locomotion behavior under the chemotactic gradient was totally different to both, the absence of stimulus, and the presence of an electric field.

**Induction process**. Once the migrations of the amoebae in the three previous basic and independent experimental conditions (without stimulus, under galvanotaxis and under chemotaxis) were analyzed, we studied the trajectories of 180 *Amoeba proteus* (experimental replicates: 32, number of cells per replicate: 4–10) when they were exposed simultaneously to galvanotactic and chemotactic stimuli for 30 min (Fig. 4a). For such a purpose, we arranged the cathode on the right of the set-up and the anode with the nFMLP peptide solution on the left. The analysis of the amoebae trajectories showed that 53% of the cells ignored the

electric field signal and moved towards the anode-peptide (23% of them exhibited a very sharp directionality), while the remaining 45.33% migrated to the cathode. Three cells (1.67%) presented an atypical behavior, remaining immobile but adhered to the substrate during the 30 min of the test, and therefore were included in the unconditioned group. The cosines of the displacement angles were distributed between −1 and 1 (−0.26/1.8 median/IQ). This analysis verified that two fundamental cellular migratory behaviors had emerged in the experimental system, one towards the anode and another towards the cathode. The statistical analysis confirmed the presence of these different behaviors ($p = 10^{-30}$; $Z = −11.435$, Wilcoxon rank-sum test). In Fig. 6, a galvanotactic control of the cells that responded to the cathode during the induction process is shown. All the amoebae again migrated towards the cathode, confirming that these cells were unconditioned.

**Conditioned behavior test**. To verify if the cells that moved towards the anode during the induction process (Fig. 4a) present some kind of persistence in their migratory behavior, we analyzed 160 amoebae in three different scenarios in which they were exposed to several types of perturbations.

In the first scenario, 85 amoebae (experimental replicates: 32, number of cells per replicate: 1–7) that had previously migrated

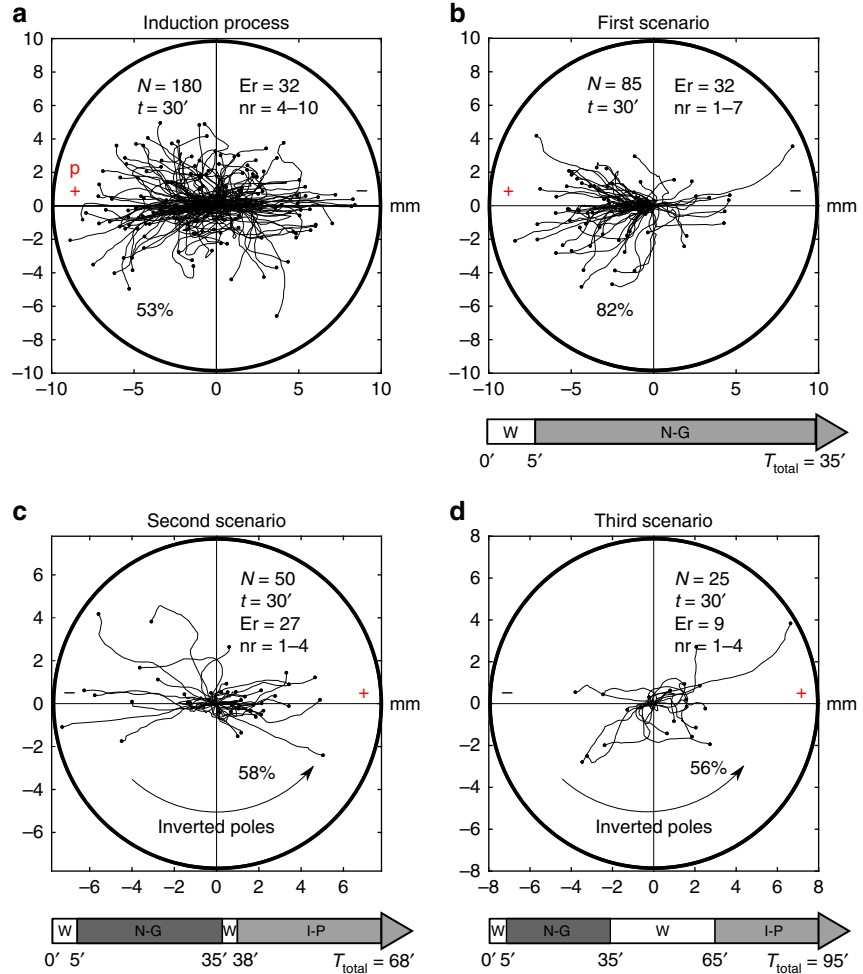

**Fig. 4** Experimental evidences of conditioned behavior in *Amoeba proteus*. **a** Under galvanotaxis and chemotaxis simultaneously, 53% of the amoebae moved towards the anode-peptide (induced cells). **b** First scenario (after induction process, the cells were exposed to 5 min without stimulus and 30 min on galvanotaxis), 82% of the induced cells presented lasting directionality towards the anode where the chemotactic peptide was absent. **c** Second scenario (after induction process, the cells were exposed to 5 min without stimulus, 30 min on galvanotaxis, 3 min without stimulus and 30 min on galvanotaxis with inverted polarity), 58% of the induced cells continued migrating towards the positive pole, now in the opposite side. **d** Third scenario (after induction process, the cells were exposed to 5 min without stimulus, 30 min on galvanotaxis, 30 min without stimulus and 30 min on galvanotaxis with inverted polarity), 56% of the induced cells still maintained the migration towards the positive pole. "*N*" total number of cells, "Er" experimental replicates, "nr" number of cells per replicate, "*t*" galvanotaxis time, "*T*" total time after induction process, "p" chemotactic peptide (nFMLP), " + " anode, "−" cathode. "W" without stimulus, "N-G" normal galvanotaxis, "I-P" galvanotaxis with inverted polarity. In all the diagrams, the tracking has been represented up to the maximum value obtained towards the positive or negative pole. Both the *x* and *y* axis show the distance in mm

towards the anode-peptide during the exposition to two simultaneous stimuli (induction process) were manually extracted and placed for 5 min on a normal culture medium (Chalkley´s medium) in a small Petri dish in absence of stimuli. Then the cells, were deposited on a new identical glass and block set-up that had never been in contact with the chemotactic peptide nFMLP and exposed for the second time to a single electric field, without peptide, during 30 min (note that the total time after the first induction process was 35 min). The analysis of the individual trajectories showed that 82% of the cells ran to the anode where the peptide was absent (Fig. 4b). The cosines of displacements ranged between −1 and 0.998 (−0.854/0.77 median/IQ). This result supported mathematically that the majority of cells moved towards the anode in the absence of peptide, thus corroborating that a new locomotion pattern had appeared in the cells. Such systemic behavior (migration towards the anode in the absence of peptide) had never appeared before. The comparison between the cosines of displacements obtained during the galvanotaxis without previous induction (Fig. 3b) and the galvanotaxis after

the induction (Fig. 4b) showed that this newly acquired cellular behavior is extremely unlikely to be obtained by chance ($p = 10^{-19}$; $Z = 8.878$, Wilcoxon rank-sum test). Four cells in this scenario exhibited eventual displacements towards both sides without any preference, and therefore were included in the unconditioned group. Since 43 cells persisted in the migration towards the anode until the end of the galvanotaxis, we used 50 cells for the next step.

In the second scenario, we studied 50 amoebae (experimental replicates: 27, number of cells per replicate: 1–4) previously exposed to the induction process and to the conditioning test of the first scenario (5 min without stimulus and 30 min of galvanotaxis). After that, these cells were placed once more in Chalkley´s medium without any stimulus for 3 min, and then they were again exposed to galvanotaxis for 30 min but in this case the polarity of the electric field was inverted (the cathode was positioned where the anode was previously and vice versa). In short: after the induction process, the cells were exposed to 5 min without stimulus, 30 min on galvanotaxis, 3 min without stimulus

and 30 min on galvanotaxis with inverted polarity; in total, the time elapsed between the end of the induction process and the end of the study was 68 min.

When an *Amoeba proteus* is placed in an electric field for long periods of time, the probability of dying or, at least, detaching from the substrate and adopting a spherical shape increases sharply. Therefore, the cells were physically extracted and replated for 3 min to minimize cell damage. Next, we changed the medium in the set-up, after that the cells were again placed in the clean experimental chamber, and finally exposed to galvanotaxis with inverted polarity with fresh medium for 30 min. By inverting the electric field, we demonstrated that the amoebae were neither directed to a specific point in the space nor they associated a specific point in the space to the peptide.

Even under these new and strict conditions, 58% of the cells continued migrating towards the anode (now positioned in the opposite side) thus maintaining the conditioned behavior (Fig. 4c). The remaining cells (42%) lost this ability. The comparison between the cosines of displacements in the galvanotaxis without induction (Fig. 3b) and the cosines from the second scenario also indicated that it is unlikely to obtain this new conditioned systemic behavior by chance ($p = 10^{-11}$; $Z = -6.491$, Wilcoxon rank-sum test). Four cells displayed an atypical behavior characterized by immobility and adhesion to the substrate during the 30 min of the test, and three more cells showed eventual displacements towards both sides; all of them were included in the unconditioned group. Since 16 cells maintained the migration towards the anode until the end of the galvanotaxis, we used 25 cells for the next step.

In the third scenario, 25 cells (experimental replicates: 9, number of cells per replicate: 1–7) were exposed to the induction process and to the conditioning test of the first scenario (5 min without stimulus and 30 on galvanotaxis). Next, once again, the cells were placed in Chalkley´s medium without any stimulus, in this occasion for about 30 min, and then exposed to galvanotaxis with inverted polarity for 30 min like in the second scenario (Fig. 4d). In short, 5 min without stimulus, 30 min on galvanotaxis, 30 min without stimulus and 30 min on galvanotaxis with inverted polarity; therefore, the total time elapsed between the end of the induction process and the end of the study was 95 min.

Like in the second scenario, cells were physically extracted and re-plated for 3 min to minimize cell damage due to the long standing in an electric field. Next, we changed the medium in the set-up, then the cells were again placed in the clean experimental chamber, and finally exposed to galvanotaxis with inverted polarity with fresh medium for 30 min.

Even under these stricter conditions, 56% of the cells still maintained the migration to the anode, positioned in the opposite side, making evident that the new systemic behavior exhibits a remarkable robustness despite the perturbations introduced in the experiment. In addition, the test comparing the cosines during the third scenario (Fig. 4d) and the galvanotaxis without previous induction (Fig. 3b) showed that it is completely unlikely to obtain this newly acquired cellular behavior by chance ($p = 10^{-11}$; $Z = 6.221$, Wilcoxon rank-sum test). One cell presented an atypical behavior characterized by immobility, and therefore was included in the unconditioned group.

A very restrictive criterion was adopted in these three scenarios: a cell was considered to present lasting directionality towards the anode only if after 15 or more minutes, the amoeba was still migrating towards the anode (which corresponds to 20 min after the induction process since all the amoebae were placed in Chalkley's medium without any stimulus for 5 min after the induction) or if after moving initially towards the cathode, the cell corrected its trajectory showing a clear lasting directionality

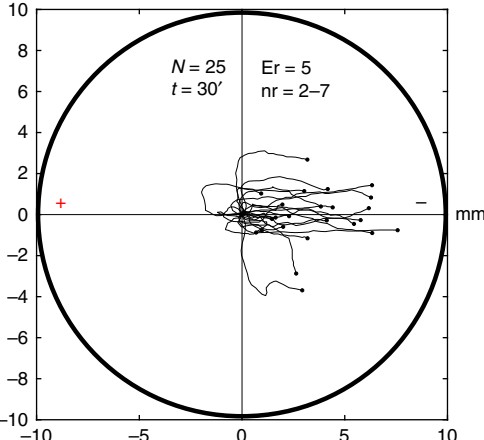

**Fig. 5** Galvanotactic control of the cells that responded to the cathode during the induction process. 25 *Amoeba proteus* (5 experimental replicates, 2–7 cells per replicate) that migrated towards the cathode during the induction process were again subjected to a controlled electric field (galvanotaxis during 30 min). All the amoebae migrated towards the cathode, confirming that these cells were unconditioned, and that their natural behavior was not altered by the induction process. "*N*" total number of cells, "Er" experimental replicates, "nr" number of cells per replicate, "*t*" galvanotaxis time, " + " anode, "−" cathode. Both the *x* and *y* axis show the distance in mm, and the initial location of each cell has been placed at the center of the diagram

towards the anode. Note that these behaviors were never observed in any of the five galvanotactic experiments (Figs. 3b, 5 and 6a–c).

**Persistence time of the new acquired cellular behavior**. The new emergent systemic behavior is also characterized by a limited duration through time. Figure 7a is an illustrative example of the loss of conditioning in 15 induced cells (experimental replicates: 4, number of cells per replicate: 3–6) as time goes on. To quantify this phenomenon, we have measured the duration time of the conditioned behavior in the 148 conditioned cells in the three scenarios previously described.

In the first scenario, (35 min after the induction process, 81 conditioned cells) the analysis of the persistence level showed that 11 cells lost the inducted behavior at the beginning of the test, 27 cells did it after 20–33 min, whereas the rest (43 cells, 53%) maintained migration towards the anode until the end of the experiment (see Fig. 7b for details).

In the second scenario, (68 min after the induction process, 43 conditioned cells) 14 cells lost the persistent behavior at the beginning of the test and 13 did it after 52–66 min, whereas the remaining 16 cells (37%) continued the migration towards the anode until the end of the experiment (Fig. 7b).

In the third scenario, (95 min after the induction process, 24 conditioned cells) 10 cells lost the persistent behavior at the beginning of the test, 6 did it after 47–90 min, and the remaining 8 cells (33%) continued the migration towards the anode until the end of the experiment (Fig. 7b).

Despite all the perturbations introduced in the experiments, the whole analysis indicates that the average time of the cells that lost the acquired motility pattern during any of the three scenarios was 44.04 ± 21.8 min.

**Evidences of associative conditioning in *M. leningradensis*.** In order to examine the robustness of the observed conditioned behavior we have performed a preliminary study on another unicellular species, *Metamoeba leningradensis*, under the same

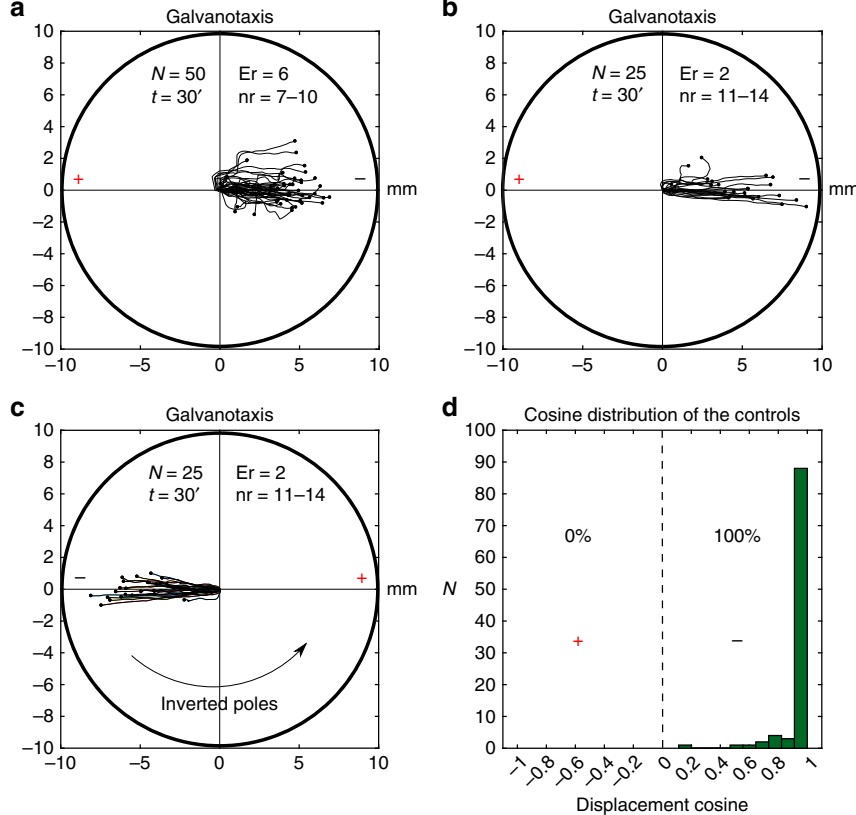

**Fig. 6** Controls of galvanotactic behavior in *Amoeba proteus*. **a** 50 amoebae (experimental replicates: 6, number of cells per replicate: 7–10) were subjected to a controlled electric field (galvanotaxis during 30 min) after being exposed to nFMLP peptide for at least 30 min. All cells exhibited normal galvanotaxis, that is, migration towards the cathode, confirming that exposure to nFMLP was not able to elicit changes in the normal galvanotactic response of *Amoeba proteus* by itself. **b** 25 amoebae (experimental replicates: 2, number of cells per replicate: 11–14) were subjected to a galvanotactic stimulus during 30 min, next, the cells were placed in a Petri dish filled with Chalkley's medium for 3 min and then exposed to another identical electric field with inverted polarity **c** (experimental replicates: 2, number of cells per replicate: 11–14). All the amoebae showed a normal galvanotactic behavior in both occasions **b**, **c**. **d** Distribution of the cosines of displacement angles for all the trajectories under galvanotaxis (**a**, **b** and **c**). In order to represent the preference towards a pole for the three controls simultaneously, the signs of the cosines of displacement angles of the trajectories represented in panel **c** were inverted (associating the anode to negative cosines and the cathode to positive values). As it was observed previously in Fig. 3e, 100% of the displacement cosines were positive, indicating a strong preference towards the cathode when the electric field was active. "N" total number of cells, "Er" experimental replicates, "nr" number of cells per replicate, "t" galvanotaxis time, "+" anode, "−" cathode. Both the x and y axis show the distance in mm, and the initial location of each cell has been placed at the center of the diagram

induction process as the *Amoeba proteus*. The metamoebae were exposed to the same intensity of the electric field and to the same peptide (nFMLP) concentration, and therefore, values of amperage or optimal concentration of peptide were not adapted to generate more efficient responses of these organisms to such stimuli.

Figure 8a shows the galvanotactic locomotion of 50 cellular trajectories (experimental replicates: 4, number of cells per replicate: 4–15) analyzed under an external, controlled direct-current electric field of about 340 mV/mm. This study indicated that practically all the metamoebae migrated towards the cathode during 30 min. The quantitative analysis showed that the values of the cosines of displacements were distributed between 0.04 and 1 (0.98/0.09 median/IQ), which verified that a unique fundamental behavior characterized by an unequivocal directionality towards the cathode had emerged in the experimental system.

Next, 160 metamoebae (experimental replicates: 15, number of cells per replicate: 3–12) were subjected to an induction process of 30 min. They were exposed simultaneously to chemotactic and galvanotactic stimuli, placing the peptide on the anode side (Fig. 9). Under these conditions, the record of the migration trajectories showed that 39% of the metamoebae ignored the

electric field signal and moved towards the anode, while the remaining (61%) migrated to the cathode. The cosines of the displacement angles were distributed between −1 and 1 (0.54/1.8 median/IQ). Hence, the result of the induction process indicated that two fundamental cellular migratory behaviors had emerged in the experimental system, one towards the anode and another towards the cathode. The Wilcoxon rank-sum test confirmed the presence of these two different behaviors ($p = 10^{-26}$; $Z = -10.639$, Wilcoxon rank-sum test).

Finally, to verify whether the cells that moved towards the anode during the induction process presented some kind of conditioning in their migratory trajectories, we performed a conditioned behavior test (Fig. 8b). For such a purpose, the 62 *Metamoeba leningradensis* (experimental replicates: 15, number of cells per replicate: 1–7) that had previously migrated towards the anode-peptide during the induction process were manually extracted and placed for 5 min into a normal culture medium (Chalkley's medium) in a small Petri dish, in absence of stimuli. Then, in a similar way that we did in the *Amoeba proteus* experiments, the metamoebae were placed usually in groups of 1–6 on a new identical glass and block set-up that had never been in contact with the chemotactic peptide. Under these conditions,

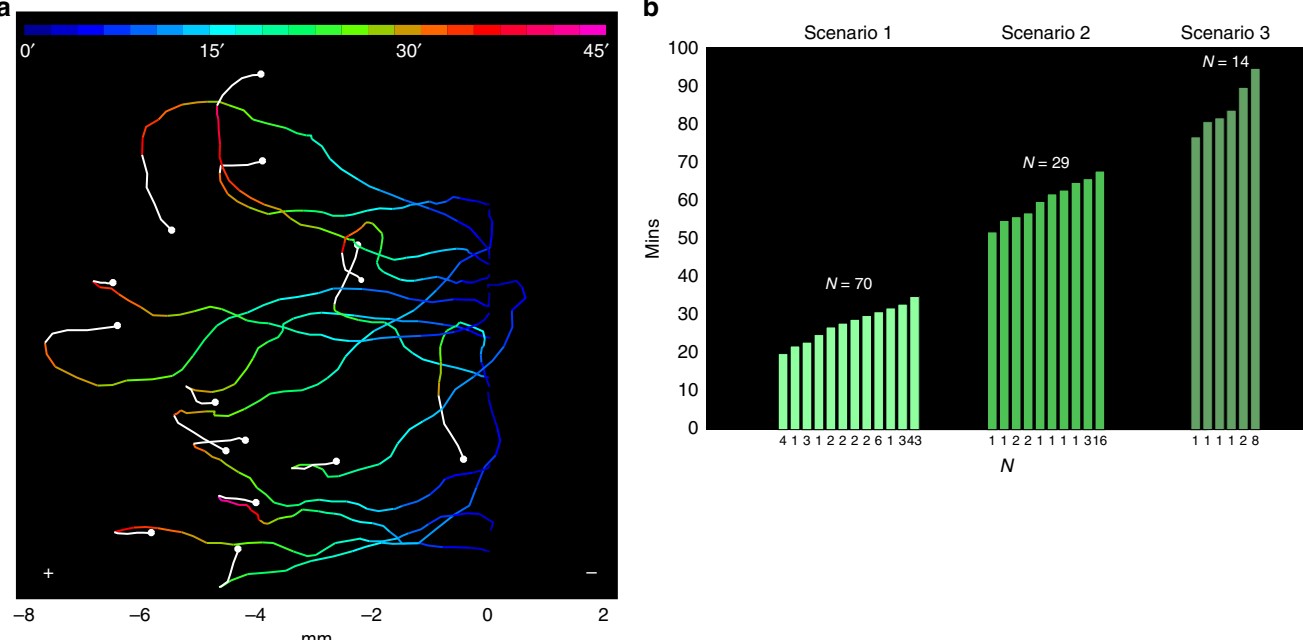

**Fig. 7** Persistence time in the conditioned motility patterns of *Amoeba proteus*. **a** Trajectories of 15 amoebae under galvanotaxis, that previously acquired the systemic conditioned behavior after induction process, lost gradually the persistence towards the anode (times ranging from 27 to 44 min) and turned back to the cathode (white line). The colors of the trajectories represent the duration of the conditioned behavior, as is indicated in the top of the figure. **b** Histogram representing the durations of the conditioned behavior for the first, second, and third scenarios. The persistence times ranged between 20 and 95 min. The cells that lost the persistent conditioned behavior at the beginning of the tests were not represented. The three scenarios were shaded in different green tones for better comprehension

they were exposed for the second time to a single electric field, without peptide, during 30 min (the total time after the induction process was 35 min).

The analysis of the individual trajectories showed that 71% of the cells ran to the anode where the peptide was absent (Fig. 8b). The cosines of displacements ranged between $-1$ and 0.99 ($-0.539/1.33$ median/IQ). This result supported mathematically that the majority of cells moved towards the anode in the absence of peptide, thus corroborating that a new locomotion pattern had appeared in the metamoebae cells. Such systemic behavior (migration towards the anode in the absence of peptide) had never appeared before. The comparison between the cosines of displacements obtained during the galvanotaxis without previous induction (Fig. 8a) and the galvanotaxis after the induction (Fig. 8b) showed that this newly acquired cellular behavior is extremely unlikely to be obtained by chance ($p = 10^{-17}$; $Z = 8.326$, Wilcoxon rank-sum test) in *Metamoeba leningradensis*.

### Discussion

Here, using an appropriate direct-current electric field (galvanotaxis) and a specific peptide (nFMLP) as a chemoattractant (chemotaxis) we have addressed essential aspects of the *Amoeba proteus* and *Metamoeba leningradensis* migration. More precisely, we have found that these cells can link two different past events, shaping an associative conditioning process characterized by the emergence of a new type of systemic motility pattern. This behavior consists in a persistent migration towards the anode when these cells typically migrate to the cathode.

First we have studied the *Amoeba proteus* migration and we have verified in the galvanotactic experiments that practically all the amoebae show an unequivocal systemic response consisting in the migration towards the cathode when they are exposed to a strong direct electric field of about 300–600 mV/mm (Fig. 3b, and

Fig. 6). However, if the amoebae are exposed simultaneously to a chemotactic and galvanotactic stimulus (induction process), placing a specific peptide in the anode, 53% of the amoebae moved to the site where the peptide was located, and most of these cells (82%) were able to acquire a new singular behavior in their systemic locomotion characterized by a persistent migration towards the anode, which was observed in subsequent galvanotaxis experiments carried out in absence of peptide (Fig. 4b–d).

This extensive study, which has covered 615 cellular trajectories, has shown that when the exposition to a stimulus related to the amoeba's nourishment (a specific peptide) is accompanied by an electric field, and the peptide is placed in the anode, the amoebae appear to associate the anode with the food (the peptide) and after the induction process most cells developed a new persistent pattern of cellular motility characterized by movements towards the anode even if the nourishment (peptide) was absent. After an induction process, most of amoebae seem to associate food with the anode and, consequently, modify their conduct, behaving against their known tendency to move to the cathode. Strikingly, this induced association of anode and food can be maintained for relatively long periods of time. In our experiments, this conditioned motility pattern prevailed for periods ranging from 20 to 95 min. This period of time is very long in comparative terms if we take into consideration that the cellular cycle of *Amoeba proteus* lasts, with small variations depending on the environment, only about 24 h under controlled culture conditions[22]. We have also observed that, after the induction process, a small subset of the amoebae was not conditioned. Cells display a range of differences in their membrane receptors, electric potential, physiological/metabolic functioning, and hence, there are no two identical unicellular organisms. In our experiments, some cells probably were unconditioned or weakly conditioned due to their intrinsic physiological peculiarities, and in addition, some kind of cellular damage caused by the experimental process may have occurred.

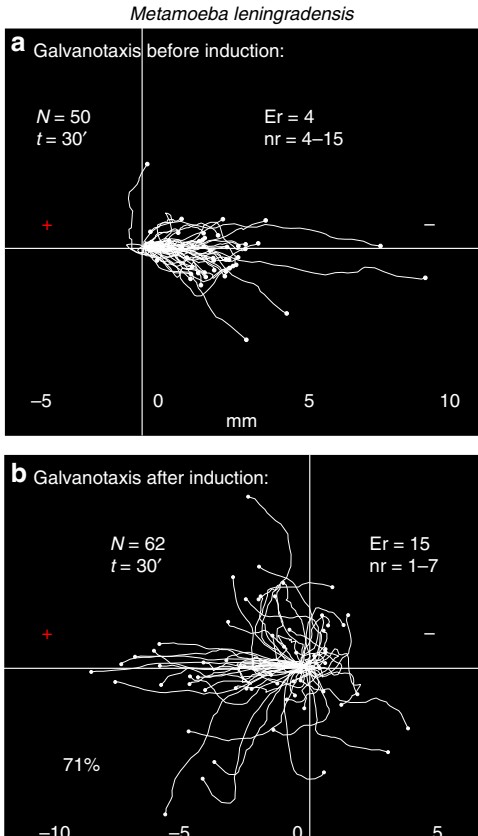

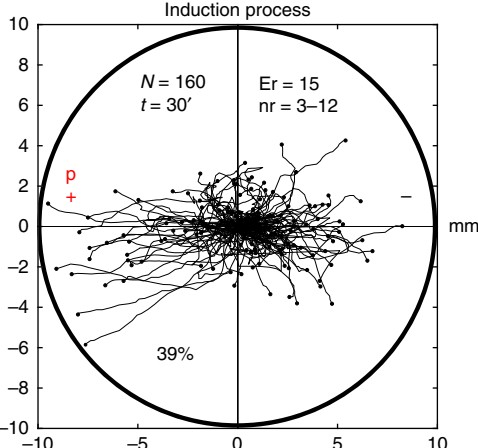

**Fig. 9** Induction process in *Metamoeba leningradensis*. Under galvanotaxis and chemotaxis simultaneously, 39% of the *Metamoeba leningradensis* moved towards the anode-peptide (induced cells). "*N*" total number of cells, "Er" experimental replicates, "nr" number of cells per replicate, "*t*" galvanotaxis time, "p" chemotactic peptide (nFMLP), "+" anode, "−" cathode. The tracking has been represented up to the maximum value obtained towards the positive or negative pole. Both the *x* and *y* axis show the distance in mm

**Fig. 8** Conditioning process in *Metamoeba leningradensis*. **a** Under galvanotaxis conditions practically all the *Metamoeba leningrandensis* migrated towards the cathode. **b** After induction process, the cells were placed in Chalkley's medium without any stimulus for 5 min, and then they were exposed to galvanotaxis for 30 min. 71% of the induced cells presented lasting directionality towards the anode, where the chemotactic peptide was absent. "*N*" total number of cells, "Er" experimental replicates, "nr" number of cells per replicate, "*t*" time of galvanotaxis or chemotaxis, "+" anode, "−" cathode. Both the *x* and *y* axis show the distance in mm, and the initial location of each cell has been placed at the center of the diagram

To test the robustness of the conditioned behavior we have performed a preliminary study in *Metamoeba leningradensis* under the same strict conditions that we set up for *Amoeba proteus*. Despite these restrictive conditions, most metamoeba cells were able to link two different past events, same as *Amoeba proteus*, shaping an associative conditioning process characterized by the emergence of a new type of systemic motility pattern which consists in a persistent migration towards the anode when, in the absence of previous induction, these cells also typically migrate to the cathode (Fig. 8).

The controls carried out during the research indicated that cells exposed independently either to galvanotaxis or chemotaxis, did not present any observable atypical behavior (Fig. 6), and the quantitative study performed emphasized that it is extremely unlikely to obtain the new type of induced systemic behavior by chance ($p = 10^{-19}$; $Z = 8.878$, Wilcoxon rank-sum test). In conclusion, the work we have performed here shows that most of the conditioned *Amoeba proteus* and *Metamoeba leningradensis* exhibited the ability to preserve the relationship between the two stimuli, acquiring a new type of systemic behavior via conditioning. Noteworthy, the fact that individual cells are able to generate associative conditioned behaviors to guide their complex migration movements has never been verified so far.

Our experimental results may allow another possible explanation. The exposure to nFMLP triggers a sub-population of cells to change the character of their migration in an electric field, making them to migrate towards the anode rather than the cathode. A notable number of cells belonging to both species can persistently change their migration pattern by these two external, simple and independent stimuli, when both are simultaneously applied. The new behavior persists for around 1 h, and gradually fades away thereafter.

Amoebae and metamoebae cells seem to associate the anode with the peptide in the induction process. After the conditioning, both stimuli seem to remain linked in these cells for a relatively long period of time, and consequently, the systemic movement of amoebae and metamoebae responded to the presence of an electric field by migrating towards the anode instead of the expected migration to the cathode.

In brief, we have observed a systemic cell behavior that can be modified by two simple external and independent stimuli, when they are simultaneously applied. This conditioned migration behavior can prevail for 44 min on average.

Pavlov studied four fundamental types of persistent behavior provoked by two stimuli. Here we have based our work in one of them, the called "simultaneous conditioning", in which both stimuli are applied at the same time.

However, in a strict sense, we cannot conclude that our findings represent the classical Pavlovian conditioning since complete controls and parametric analyses for classical conditioning studies have not been performed yet[23].

The experiments we show here were inspired by numeric predictions based on computational modeling that we published in 2013 dealing with complex metabolic networks[24]. Thus, analyzing complex enzymatic processes under systemic conditions using Statistical Mechanic tools and advanced Computational and Artificial Intelligence techniques, we were able to verify numerically that self-organized enzymatic activities in modular metabolic networks seem to be governed by Hopfield-like attractor dynamics similar to what happens in neural networks[24]. A key attribute of the analyzed metabolic Hopfield-like dynamics is the presence of associative memory. This quantitative study showed

that the associative memory in unicellular organisms is possible[24,25]. Such memory would be a manifestation of emergent properties underlying the complex dynamics of the systemic cellular metabolic networks.

It is still too early to delineate the molecular mechanisms supporting this cellular associative conditioning. However, there are evidences of a functional memory, which can be embedded in multiple stable molecular marks during epigenetic processes[25]. Likewise, long-term correlations (mimicking short-term memory in neuronal systems) have also been analyzed in experimental calcium-activated chloride fluxes in *Xenopus laevis* oocytes[26]. On the other hand, different studies have described several molecular processes in which both prokaryotic and eukaryotic cells show chemotactic memory. For instance, changing dynamics in specific methylation-demethylation patterns in prokaryotes seem to be involved in molecular memory processes related to chemical gradient adaptation[27–30]. Besides, phosphotransfer processes and other post-translational modifications seem to be involved in chemostatic cellular persistence of eukaryotic cells[31–33].

In this paper, we have addressed essential aspects of the *Amoeba proteus* and *Metamoeba leningradensis* migration. The mechanisms underlying amoeba locomotion are extremely complex and the ability to direct their movement and growth in response to external stimuli is of critical significance for its functionality; in fact, cellular life would be impossible without regulated motility. Although some progress is being made in the understanding of cellular locomotion, how cells move efficiently through diverse environments, and migrate in the presence of complex cues, is an important unresolved issue in contemporary biology. Free cells need to regulate their locomotion movements in order to accomplish critical activities like locating food and avoiding predators or adverse conditions. In the same way, cellular migration is required in multicellular organisms for a plethora of fundamental physiological processes such as embryogenesis, organogenesis and immune responses. In fact, deregulated human cellular migration is involved in important diseases such as immunodeficiencies and cancer[11,34]. Neoplastic progression (invasion and metastases), for example, can be regarded as a process in which the survival of tumor cells depends also on their ability to migrate to obtain additional resources in a general context of scarcity[35].

Here, we have verified that two unicellular organisms such as *Amoeba proteus* and *Metamoeba leningradensis* are able to modify their systemic response to a determined external stimulus exclusively by associative conditioning. This fact opens up a new framework in the understanding of the mechanisms that underlie the complex systemic behavior involved in cellular migration and in the adaptive capacity of cells to the external medium.

## Methods

**Cell cultures.** *Amoeba proteus* (Carolina Biological Supply Company, Burlington, NC.Item # 131306) were grown at 21 °C on Simplified Chalkley's Medium (NaCl, 1.4 mM; KCl, 0.026 mM; CaCl₂, 0.01 mM), alongside *Chilomonas* as food organisms (Carolina Biological Supply Company Item #131734) and baked wheat corns.

*Metamoeba leningradensis* (Culture Collection of Algae and Protozoa, Oban, Scotland, UK, CCAP catalog number 1503/6). They were cultured in the same conditions as *Amoeba proteus*.

**Experimental set-up.** All the experiments were performed in a specific set-up (Fig. 1) consisting in two standard electrophoresis blocks, 17.5 cm long (Biorad Mini-Sub cell GT), a power supply (Biorad powerbank s2000), two agar bridges (2% agar in 0.5 N KCl, 10–12 cm long) and a structure made from a standard glass slide and covers commonly used in Cytology Laboratories.

The first electrophoresis block was directly plugged into the power supply while the other was connected to the first via the two agar bridges, which allowed the current to pass through and prevented the direct contact between the anode and cathode and the medium where the cells would be placed later. Both electrophoresis blocks consisted of 3 parts: on the extremes, there are

2 wells which were filled by the conductive medium (Chalkley's Simplified Medium) and, in the middle, an elevated platform (Fig. 1).

In the center of the second electrophoresis block we placed the experimental chamber that allowed us to obtain a laminar flux when it was closed and the addition and extraction of cells when it was open.

The experimental chamber consisted in 4 pieces of glass (standard glass slide and covers commonly used in Cytology Laboratories), a 75 × 25 mm modified slide and three small pieces obtained by trimming of three cover glass of 60 × 24 × 0.1 mm (Fig. 1).

Three cover glasses were trimmed with a methacrylate ruler, one measuring about 3 × 24 × 0.1 mm and two measuring about 40 × 24 × 0.1 mm each, here on called central piece and sliding lateral pieces, respectively. These three glasses were for only one use.

This glass structure (Fig. 1) supported the sliding parts of the experimental chamber. It was reusable after cleaning. To build it, we fixed with silicone on a glass slide along the two longest edges of the slide (if the width of a cover is 24 mm, about 4 mm were stuck on the slide and 20 mm protrude towards the outside of the glass slide). Then we left it to dry for 24 h. The last step consisted on trimming the protruding portions of the cover slides (about 60 × 20 × 0.1 mm) with a methacrylate ruler, so that two small longitudinal strips of approximately 60 × 4 × 0.1 mm were adhered to the glass slide (see Fig. 1), which shaped the lateral limits of the experimentation chamber.

The modified slide was placed in the central platform of the second electrophoresis block. To avoid medium going across the modified slide from below, we placed an oil drop in the central platform of the block of electrophoresis, on which the modified slide was placed. It is very important that the oil drop expands to cover the entire width of the experimental chamber. In the center of the modified slide, without any glue, we placed the central piece and the two sliding lateral pieces leaving short distance between all of them.

The amoebae were placed below the central piece of the chamber in an approximate volume of 30 µl. To note, it is crucial that the amoebae do not remain for more than a few seconds in the micropipette tip to avoid the adhesion of the amoeba to the inner surface of the tip.

Once the amoebae were placed under the central piece of the chamber, we waited for two minutes to allow the cells to stick to the surface of the modified slide. Then, we filled the wells of the electrophoresis blocks with simplified Chalkley medium up to the level necessary to contact with the base of the modified slide, but not the two sliding lateral pieces. Later, the two sliding lateral pieces were pushed with two micropipette tips until they contact with the liquid in the wells. Next, the two sliding lateral pieces are pushed to contact the central piece in the chamber. This way, a laminar flux can be established throughout the inner space of the experimental chamber.

In the induction process, once the laminar flux was created, and before the activation of the electric field, we added 750 µl of 2 × 10⁻⁴ M nFMLP peptide solution to the medium (75 ml) in the positive pole well of the second electrophoresis block.

Considering that the amoebae that had shown a specific behavior were needed to perform further experiments, the cells were collected opening the sliding lateral pieces with the tip of a micropipette.

Set of videos intended only for didactical purposes. They are merely descriptive, to make easier the understanding of our experimental procedure and the reproducibility of our studies. Note that steps 5 to 8 are performed directly under the microscope, and we have not filmed them under the microscope for better visualization.

In summary, the experimental chamber consisted in a sliding glass structure. The sliding lateral pieces could be displaced in the longitudinal direction. This way, when the sliding pieces were closed an inner laminar flux was available in the chamber and, when they were open, the placement and collecting of the cells were possible easily. Movies showing the main experimental procedures have been deposited in figshare (https://doi.org/10.6084/m9.figshare.8868326).

*A. proteus* and *M. leningradensis* may display some physiological variations depending on culture conditions. Before the experiments, the cells were starved for 24 h in Chalkley's Simplified Medium (the same medium that was used in the experiments), in the absence of external stimuli. Once starved, only the cells that were strongly attached to the substrate, actively moving through it and showing a little amount of thin pseudopodia were used in the experiments.

The cells were washed in clean Chalkley's medium and placed in the middle of the glass set-up (experimental chamber), under the central piece of cover glass and left to rest until all of them appeared to be firmly attached to the bottom of the modified glass slide. Next, the two 4 cm long cover glasses (sliding lateral pieces) were placed on the sides of the glass structure, protruding outside of the middle platform of the block and over the lateral wells (Fig. 1). After that, each well was filled using 75 ml of Chalkley's medium, in such a way that the glass protrusion over each well is in contact with the liquid's surface. Finally, as the Chalkley's medium slowly filled up the experimental chamber, both lateral cover glasses had to be gently pushed towards each other until they touched the middle cover glass, completely covering the whole structure and forming a laminar flux that connected both lateral wells.

The experiments were always made with small groups of cells. For instance, in *Amoeba proteus* along the induction process, we analyzed a total of 180 cells that were studied in 32 different times (experimental replicates) analyzing them in groups of 4–10 cells each (number of cells per replicate). Scenario 1 was repeated 32 times. Scenarios 2 and 3 were performed 27 and 9 times, respectively.

The induction process was usually performed using around 7 cells per experiment, sometimes as few as 4 or 5 and other times as many as 9 or 10, the average being 6–8 cells. The number of cells analyzed in the scenarios depended on how many cells appeared to be conditioned in the first step, so that the number of cells per experiment is lower each time, for instance, in scenario 1, the number of cells was usually between 2 and 4. Finally, in scenarios 2 and 3 the experiments were performed using fewer amounts of cells per experiment, usually 1–3 which were the cells that migrated towards the cathode during the conditioning process.

Compared to *Amoeba proteus*, the *Metamoeba leningradensis* showed a more varied array of behaviors and shapes. These cells were also more difficult to handle, as they were more prone to strongly stick to the micropipette tips, while usually showing a weaker attachment to the glass chambers.

**Electric field (galvanotaxis).** An electric field was applied to the first electrophoresis block, which was then conducted to the second by the two agar bridges. Direct measurements taken with a multimeter in the second block (where the cells were placed) showed that the strength of the electric current oscillated between 58.5 and 60 V (334–342 mV/mm) while the intensity values varied between 0.09 and 0.13 mA.

After 30 min of exposure, during which the cellular migration movement were recorded, the power supply was turned off and the agar bridges removed.

All the experiments where the only stimulus was an electric field were performed in an electrophoresis block that had never been in contact with any chemotactic substance.

**Cell induction.** Groups of 4 to 10 amoeba were placed in the experimental chamber. Once all the cells were attached to the glass surface, the laminar flux was stablished by gently closing the structure using the sliding cover glasses. Next, the peptide, nFMLP was introduced in the left well of the electrophoresis block. After about 1 min, the power supply was turned on and the electric field established. The process lasted for 30 min; after that, the power supply was turned off and the cells that moved towards the anode removed from the experimental chamber and placed in a Petri dish with clean Chalkley's medium for future experiments (Scenario 1, 2 or 3).

When the cells were subjected to both stimuli at the same time (induction process), a new population response arose. This population behavior, in *Amoeba proteus*, showed that about half the amoebae cells migrated towards the anode (where the nFMLP peptide was placed), and approximately the other half of the cellular population migrated to the cathode (Fig. 4). On the other hand, only 39% of the *Metamoeba leningradensis* moved towards the anode-peptide, in the induction process (Figure S1). Accordingly, the cellular migration response under two simultaneous stimuli is notoriously different from that observed when the stimuli were separated (Figs. 3b, c and 8a), and therefore it cannot be concluded that the chemotactic gradient has a stronger influence on cell migration than the electric field.

In order to homogenize the cellular responses as much as possible, we put all the amoebae under starving conditions for at least 24 h before performing any experiments.

**Peptide gradient (chemotaxis).** Once the laminar flux was created, we added 750 µl of $2 \times 10^{-4}$ M nFMLP (#F3506, Sigma-Aldrich) peptide solution to the medium (75 ml) in the positive pole well of the second electrophoresis block; therefore, the peptide solution was diluted to a final concentration of $2 \times 10^{-6}$ M. In all our experiments, we used the same concentration of nFMLP. In order to homogenize the solution and accelerate the creation of a chemotactic gradient in the experimental chamber we carefully mixed the content of the left well until the amoebae appeared to start moving towards it. Finally, the cells behavior was recorded for 30 min.

**Peptide gradient calculation.** The generation of an nFMLP peptide gradient was evaluated by the measurement of its concentration in the middle of the experimental chamber. To this end, 4µM fluorescein-tagged peptide (#F1314, Invitrogen), was loaded in the left side of the set-up (Fig. 1). Next, the central glass piece of the experimental chamber was slightly displaced and a small opening, the size of the tip of a 50–200 µL micropipette, was left between the sliding cover glasses and the central glass piece. This little separation allowed us to get samples of 60 µL from the middle part of the laminar chamber flux at 0, 2, 5, 10, 15, 20 and 30 min following the establishment of the laminar flux. Peptide concentration was calculated extrapolating the values from a standard curve with known concentration of the fluorescein-tagged peptide (Fig. 2). All measurements were duplicated and the experiment was repeated three times. Fluorescence was measured in 96 well glass bottom black plates (P96-1.5H-N, In Vitro Scientific) employing a SynergyHTX plate reader (Biotek) at Excitation/Emission wavelengths of 460/528 following standard laboratory techniques as described by Green and Sambrook[36].

**Track recording and digitizing.** The motility of the cells was recorded using a digital camera attached to a SM-2T stereomicroscope. Images were acquired every 10 seconds, over a period of at least 30 min (180 frames). If a video was longer, only the first 30 min were quantified, except for the ones used for Fig. 7a. Since

automated tracking software is often inaccurate[37], we performed manual tracking using the TrackMate software in ImageJ (http://fiji.sc/TrackMate)[38], as suggested in by Hilsenbeck et al.[37] Each track corresponds to an individual amoeba.

**Directionality analysis and statistical significance.** In order to quantify and compare the directionality of cell migration towards the anode or the cathode, we computed the cosines of the angles of displacement of each amoeba[17]. More precisely, we calculated the cosine of the angle formed between the start and final positions of each cell. Consequently, we were able to analyze quantitatively if an amoeba moved towards the cathode (positive values of the cosine), or towards the anode (negative values). In addition, this study suggested the degree of directionality, since values closer to 1 (or to −1 in the case of the anode) indicated a very high preference towards that pole. Next, to estimate the significance of our results, we studied first if the distribution of cosines of angles came from a normal distribution, by applying the Kolmogorov-Smirnov test for single samples. Since the normality was rejected, the groups of cosines were compared in pairs by a non-parametric test, the Wilcoxon rank-sum test, and therefore, the results were depicted as median/IQ instead of as mean ± SD. Besides the p-value, we have reported the Z-statistic of the Wilcoxon rank-sum test[39].

Note that the signs of the cosines from the second and third scenarios were changed to perform the respective tests with the galvanotaxis without previous induction (Fig. 3b) because the polarity of the electric field was inverted.

Researchers involved in the quantitative analysis of the cellular trajectories were never aware of what scenario each trajectory belonged to. Only when all the trajectories were quantified and processed, the researchers in charge of recording the amoeba's movements informed the rest of the team about which trajectories belonged to each experiment or control.

**Reporting Summary.** Further information on research design is available in the Nature Research Reporting Summary linked to this article.

## Data availability

All original videos obtained in the experiments can be found in figshare, with https://doi.org/10.6084/m9.figshare.8241284 (https://figshare.com/s/c59323fabced0c533fae). In addition, movies showing the main experimental procedures can be found in figshare with the https://doi.org/10.6084/m9.figshare.8868326 (https://figshare.com/articles/Set-Up_Video_Files/8868326).

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

## Acknowledgements

We would like to thank Dr. Andrew Goodkov from the Institute of Cytology (Russian Academy of Science) St. Petersburg, Russia, for valuable advices related to Amoeba organisms, Laura Pérez Gómez and Luis Rojo García for their assistance designing Fig. 1 and the AutoCAD 3D model, A-M Pérez Biedermann for her valuable contribution in our study, José González Romero and José Miguel Pérez Pérez from the Institute of Parasitology and Biomedicine "Lopez-Neyra" for their technical assistance. In addition, we thank María Calleja-Felipe for her valuable help in the peptide gradient experiments. This work was supported by a grant of the University of Basque Country (UPV/EHU), GIU17/066, the Basque Government grant IT974-16, and by the UPV/EHU and Basque Center of Applied Mathematics, grant US18/21".

## Author contributions

C.B., M.F. and J.C.-P.: performed the experiments; C.B.: performed the digitalization of trajectories; I.M. and L.M.: performed the quantitative studies; S.K. and M.M.: performed the gradient analysis; M.D.B. main funding and laboratory facilities; J.I.L.: supplementary movies; L.M., M.D.B., A.P.S., G.P.Y., J.I.L. and I.M.: designed the research mapping; all authors wrote the manuscript and agreed with its submission; I.M.D.F.: conceived, designed and directed the investigation.
