## [Peer Review File · Nature Communications]

Reviewers' comments:

Reviewer #1 (Remarks to the Author):

In the manuscript "Evidences of Pavlovian associative conditioning in *Amoeba proteus*" the authors report a study of migration of *Amoeba proteus* in electric field and under gradients of nFMLP, which is acting as a chemoattractant, and arrive at the conclusion that *Amoeba proteus* "is able to modify its systemic response to a specific stimulus by Pavlovian associative conditioning." This reviewer does not find this conclusion to be convincingly supported by the experimental data presented in the manuscript. In addition, the descriptions of the experimental setup, methods, and procedures are not sufficiently detailed and are often unclear or confusing, complicating the evaluation of the results reported in the manuscript.

The key experiments that are supposed to support the conclusion about the Pavlovian conditioning appear to be described in a segment between lines 284 and 304 as the "the first scenario". It is the understanding of this reviewer that in these experiments cells were first exposed to a gradient of nFMLP and an electric field, which, when applied separately, would cause cells to migrate in opposite directions. Then, after 30 min, cells that migrated towards the anode were selectively extracted, placed in a medium without nFMLP for 5 min, and then exposed again to an electric field, but now without nFMLP gradient. At this point, the majority of cells migrated towards the anode, whereas naïve cells always migrate towards the cathode. (At least, this is how this reviewer understood the experiments reported in this segment.) Whereas these results are intriguing, they raise multiple immediate questions, which are not addressed in the text.

(1) 5 minutes is a short time. It is hard to see how, within this time, multiple cells could be reliably selected based on whether they were migrating towards the anode, then detached from the substrate, and then transferred to a separate Petri dish with a different medium, and then moved to some other chamber with an electric field, and allowed to attach to the substrate over there, all of that without these cells being damaged or significantly disturbed. There is no description of the transfer procedure in the text, and the description of the experimental setup is unclear. In particular, it is not clear, whether there is some chemotaxis chamber, which is closed from above (the common experimental design, which would complicate and prolong the transfer of cells), or whether cells are exposed to the gradient in some open area.

(2) It appears cells are allowed to migrate in a region with a length of at least 20 mm. It is very hard to imagine how a gradient of nFMLP that is capable of triggering chemotaxis can be reliably established in a region that long, and the manuscript does not provide any characterization of the gradient and does not provide much detail otherwise.

(3) There is no control experiment with cells that move towards the cathode during the first 30 min, whereas those cells constitute nearly 50% of the population. In addition, the sub-population of cells migrating towards the anode appears to include those that moved >7 mm as well as those that moved <1 mm. One would generally expect different outcomes for these cells, after they are extracted and placed into an electric field.

The description of the experiments in the segment between lines 306 and 351 (second and third scenario) is not completely clear. Line 308: "these cells were placed once more in Chalkley's medium without any stimulus for 3 minutes". Were these cells physically extracted and replated again? Was the medium exchanged? Or was it just simply that the electric field was turned off for 3 min and then turned on with the inverse polarity? If they were extracted and replated, then the inversion of polarity would not be meaningful. (In this respect, the caption to Fig. 5B is confusing: if cells were extracted and replated, they cannot possibly "remember" the direction of the electric field, and hence, the switching of the direction of the field would not be meaningful.) If cells stayed in place, why was the polarity not inverted without the 3 min break and why the trajectories of cells reversing their direction of migration are not shown?

Altogether, the experimental results, even when taken at face value, allow the following alternative interpretation. The exposure to nFMLP makes a sub-population of cells change the character of their migration in an electric field, making them migrate towards the anode rather than the cathode. The control in Fig. 5A does not necessarily rule it out, because in this control, cells were exposed to a single concentration of nFMLP of 0.2 nM, whereas it is not clear to what concentrations of nFMLP cells were exposed in the gradient experiment (induction). This change in the character of migration then persists for ~ 1hr, gradually fading away. While being an interesting phenomenon, it does not fit the narrative of Pavlovian associative conditioning.

Some secondary remarks and recaps. The drawings and the description of the experimental setup are unclear. The number labeling is ambiguous and dimensions are not shown in the drawing. There is no characterization of the gradient of nFMLP and of the levels of nFMLP that different cells are exposed to during the "induction". The consistency and repeatability of the gradient between different experiments is not addressed. This issue is particularly relevant, because a stable gradient stretching over 20 mm would be practically impossible to achieve. The procedures of the extraction and transfer of cells and of their placement in different media are not described.

Reviewer #2 (Remarks to the Author):

In their paper, De la Fuente et al use *Amoeba proteus* to suggest that associative conditioned behavior is present in unicellular organisms. Their findings are significant in that they provide the first evidence linking this complex process to a unicellular organism. Following Pavlov's experiments, the authors assessed the conditioning of *Amoeba proteus* by utilizing the organism's ability to migrate towards an electric field and chemical signal (i.e., peptide).

My main concerns refer to the number of biological replicates that were analyzed and whether this conditioned behavior is observed in all unicellular organisms. These concerns are detailed below.

Major points:

1. It is not clear from the manuscript how many times the experiments were repeated. For example, in Lines 260-275, 180 cells were imaged, but was this experiment repeated using cells from another cell culture? I am also somewhat concerned about the numbers of cells that were analyzed in Figs. 3C and 3D (i.e., is 25-50 cells sufficient for this type of analysis?). Furthermore, representing each cell as an experimental replicate (i.e., N) is not appropriate as there may be differences between cells from different cultures and this biological variation must be accounted for.
2. Are these observations unique to *Amoeba proteus*? To my knowledge, other unicellular amoeba do not exhibit this behavior. Given the potential significance of their conclusions, I strongly feel that the author's need to repeat these experiments using another unicellular model. This is not an unreasonable request given the potential significance of the findings and the fact that the authors have already established the experimental setup.
3. Is it possible to repeat these experiments using human cells?
4. In each of the scenarios, why are some of the cells not conditioned? The authors currently provide no insight into this observation.
5. The authors should include data showing that a gradient of peptide was established during their induction scenario. Perhaps they could label the peptide with a fluorescent molecule such as FITC?

6. Line 382: The authors need to clarify what they mean by “valid cells”. All cells in an experiment are valid (including outliers) and should be included in the analysis. They should not be omitted simply because they show something different than the average cell.

Minor points:

7. Some details about the organism (i.e., *Amoebae proteus*) need to be provided in the Introduction to set the stage for why this organism was used in the study (i.e., why was this organism the best choice for this study?).

8. Line 158: “that enabled to create” should be replaced with “enabled the creation of”

9. Lines 266-268: Is it absolutely necessary to express the percentages with 1 decimal point? I don't know if this adds anything meaningful.

10. For clarity, the data in Figure 2 should be presented as follows:

- A. Trace – without stimulus
- B. Plot of displacement angle for A
- C. Trace – galvanotaxis
- D. Plot of displacement angle for C
- E. Trace – chemotaxis
- F. Plot of displacement angle for E

11. Line 288: “located” should be replaced with “deposited”

12. Line 431: The first letter in “Petri dish” is always capitalized. Please ensure consistency in the text of the manuscript.

13. Line 476-478: The statement here should be moved to the Introduction and expanded upon (see Comment 7).

Reviewer #3 (Remarks to the Author):

This is an interesting manuscript that describes the modification of amoeba galvanotaxis produced by simultaneous pairing of an electric field with a chemical attractant (i.e., the peptide nFMLP). Although the phenomenon is interesting, the result is unlikely to have a broad impact and the manuscript suffers from conceptual, methodological and stylistic weaknesses. Specific concerns include:

1. The use of an electric field as the conditioned stimulus seems highly artificial and of little relevance to the natural behavior of amoeba.
2. Common components of classical conditioning studies include analyses of random or unpaired CS and US presentations, and sensitivity of the magnitude of the conditioning to the timing of the CS and US (i.e., forward and backward pairings). The present study is incomplete without including analyses of these key parametric features of classical conditioning.
3. Replications of the experimental groups should be provided.
4. There is no mention of blind procedures, which are essential in these studies.
5. No mechanistic analyses are presented. Consequently, the results will contribute little to the understanding of memory mechanisms. The development of another simple system with which to analyze memory mechanisms is unlikely to be of great value to the field.
6. There is excessive nonstandard terminology to describe learning phenomena and procedures (e.g., “non-conditioned stimulus,” “loss of persistence,” “double stimulus”).

7. There is nonstandard terminology to describe statistical tests (e.g., line 216, the behaviors “were absolutely different”).

Response to Reviewer # 1:

Technical Comments:

From here on, the reviewer's concerns are colored in blue, while our responses are colored in black.

(1) 5 minutes is a short time. It is hard to see how, within this time, multiple cells could be reliably selected based on whether they were migrating towards the anode, then detached from the substrate, and then transferred to a separate Petri dish with a different medium, and then moved to some other chamber with an electric field, and allowed to attach to the substrate over there, all of that without these cells being damaged or significantly disturbed. There is no description of the transfer procedure in the text, and the description of the experimental setup is unclear. In particular, it is not clear, whether there is some chemotaxis chamber, which is closed from above (the common experimental design, which would complicate and prolong the transfer of cells), or whether cells are exposed to the gradient in some open area.

(1.1) 5 minutes is a short time. It is hard to see how, within this time, multiple cells could be reliably selected based on whether they were migrating towards the anode, then detached from the substrate, and then transferred to a separate Petri dish with a different medium, and then moved to some other chamber with an electric field, and allowed to attach to the substrate over there, all of that without these cells being damaged or significantly disturbed.

The experimental chamber consists of several glass pieces with sliding capacities, allowing the laminar flux to pass throughout when the chamber is closed and the addition and extraction of cells when it is open. The chamber opens and closes easily, so the manipulation is quick.

The experiments were always made with small groups of cells. For instance, along the induction process we observed a total of 180 amoebae that were studied in 32 different experiments analyzing them in groups of 1 to 11 cells each.

The small number of cells in each group, together with the sliding structure of the experimental chamber allowed us to perform the experiments quite quickly. A new text explaining the experimental procedure has been added in the manuscript (see below).

(1.2) there is no description of the transfer procedure in the text, and the description of the experimental setup is unclear. In particular, it is not clear, whether there is some chemotaxis chamber, which is closed from above (the common experimental design, which would complicate and prolong the transfer of cells), or whether cells are exposed to the gradient in some open area.

The reviewer is right. I would like to apologize for the insufficient description of both, the experimental chamber and the experimental procedure itself. Following the reviewer's suggestion, we have included a detailed explanation of this issue in the Methods section. Additionally, we have included a set of 9 movies as supplementary material describing all the steps of the experimental procedure.

Next, we indicate the text that we have included in the Methods section.

Experimental set-up

All the experiments were performed in a specific set-up (Fig. 1) consisting in two standard electrophoresis blocks, 17.5 cm long (Biorad Mini-Sub cell GT), a power supply (Biorad powerbank s2000), two agar bridges (2% agar in 0.5 N KCl, 10-12cm long) and a structure made from a standard glass slide and covers commonly used in Cytology Laboratories.

The first electrophoresis block was directly plugged into the power supply while the other was connected to the first via the two agar bridges, which allowed the current to pass through and prevented the direct contact between the anode and cathode and the medium where the cells would be placed later. Both electrophoresis blocks consisted of 3 parts: on the extremes, there are 2 wells 5.6 cm deep which were filled by the conductive medium (Chalkley's Simplified Medium) and, in the middle, an elevated platform about 5 cm tall.

In the center of the second electrophoresis block we placed the experimental chamber that allowed us to obtain a laminar flux when it was closed and the addition and extraction of cells when it was open.

Experimental chamber

The experimental chamber consisted in 4 pieces of glass (standard glass slide and covers commonly used in Cytology Laboratories), a 75x25 mm modified slide and three small pieces obtained by trimming of three cover glass of 60x24x0.1 mm (Fig. 1).

1. Sliding parts preparation

Three cover glasses were trimmed with a methacrylate ruler (see supplementary video S1), one measuring about 3x24x0.1 mm and two measuring about 40x24x0.1 mm each, here on called central piece and sliding lateral pieces, respectively. These three glasses were for only one use.

2. Modified glass slide

This glass structure (Fig. 1) supported the sliding parts of the experimental chamber. It was reusable after cleaning. To build it, we fixed with silicone on a glass slide along the two longest sides of the slide (if the width of a cover is 24 mm, about 4 mm were stuck on the slide and 20 mm protrude towards the outside of the glass slide). Then we left it to dry for 24 hours. The last step consisted on trimming the protruding portions of the cover slides (about 60x20x0.1mm) with a methacrylate ruler, so that two small longitudinal strips of approximately 60x4x0.1 mm were adhered to the glass slide (see Fig. 1), which shaped the lateral limits of the experimentation chamber (supplementary video S2).

3. Mounting of experimental chamber on the set-up

The modified slide was placed in the central platform of the second electrophoresis block. To avoid medium going across the modified slide from below, we placed an oil drop in the central platform of the block of electrophoresis, on which the modified slide was placed. It is very important that the oil drop expands to cover the entire width of the experimental chamber. In the center of the modified slide, without any glue, we placed the central piece and the two sliding lateral pieces leaving short distance between all of them (see supplementary videos S3-S4).

4. Placement of cells in the experimental chamber

The amoebae were placed below the central piece of the chamber in an approximate volume of 30 μ l. To note, it is crucial that the amoebae do not remain for more than a few seconds in the micropipette tip to avoid the adhesion of the amoeba to the inner surface of the tip (see supplementary video S5).

5. Laminar flux establishment through the inner space of the experimental chamber

Once the amoebae were placed under the central piece of the chamber, we waited for two minutes to allow the cells to stick to the surface of the modified slide. Then, we filled the wells of the electrophoresis blocks with simplified Chalkley medium up to the level necessary to contact with the base of the modified slide, but not the two sliding lateral pieces. Later, the two sliding lateral pieces were pushed with two micropipette tips until they contact with the liquid in the wells. Next, the two sliding lateral pieces are pushed to contact the central piece in the chamber. This way, a laminar flux can be established throughout the inner space of the experimental chamber (see supplementary video S6).

6. Peptide addition to the set-up

In the induction process, once the laminar flux was created, and before the activation of the electric field, we added 750 μ l of 2×10^{-4} M nFMLP peptide solution to the medium (75 ml) in the positive pole well of the second block of electrophoresis (see supplementary video S7).

7. Cell extraction

Considering that the amoebae that had shown a specific behavior were needed to perform further experiments, the cells were collected opening the sliding lateral pieces with the tip of a micropipette (see supplementary video S8).

8. Supplementary movies of the set-up

Set of videos intended only for didactical purposes. They are merely descriptive, to make easier the understanding of our experimental procedure and the reproducibility of our studies. Note that the steps 5 to 8 are performed directly under the microscope, and we have not filmed them under the microscope for better visualization (Supplementary video S9).

In summary, the experimental chamber consisted in a sliding glass structure. The sliding lateral pieces described in step 1 could be displaced in the longitudinal direction. This way, when the sliding pieces were closed an inner laminar flux was available in the chamber and, when they were open, the placement and collecting of the cells were possible easily.

Likewise, we have included a detailed explanation in “Cell preparation” (Methods section).

(2) It appears cells are allowed to migrate in a region with a length of at least 20 mm. It is very hard to imagine how a gradient of nFLMP that is capable of triggering chemotaxis can be reliably established in a region that long, and the manuscript does not provide any characterization of the gradient and does not provide much detail otherwise.

We have confirmed the establishment of the peptide gradient by the direct measurement of fluorescein-tagged peptide concentration with a plate reader (new Fig. 2). The concentration in the middle part of the glass experimental chamber (where the amoebae are placed) increases immediately following the laminar flow establishment (within 2

minutes the concentration rises from zero to approximately 0.2 μM) and this concentration increases further with time (to 0.6 μM) for at least 30 minutes. We have addressed this issue in a new graphic in the Fig. 2 with an explanatory text in the Results and the Methods sections.

On the other hand, once the peptide was placed in the corresponding well, the amoebas in the center of the experimental chamber visibly began to respond (about 90 or 120 seconds, from the peptide is added) by changing their behavior.

(3) There is no control experiment with cells that move towards the cathode during the first 30 min, whereas those cells constitute nearly 50% of the population. In addition, the sub-population of cells migrating towards the anode appears to include those that moved >7 mm as well as those that moved <1 mm. One would generally expect different outcomes for these cells, after they are extracted and placed into an electric field.

3.1 There is no control experiment with cells that move towards the cathode during the first 30 min, whereas those cells constitute nearly 50% of the population.

We have performed the control suggested by the reviewer, and 100% of the amoebas run to the cathode. We have included a new paragraph with detailed explanations of this control in the Results and in a new Figure (Fig. 7).

3.2 the sub-population of cells migrating towards the anode appears to include those that moved >7 mm as well as those that moved <1 mm. One would generally expect different outcomes for these cells, after they are extracted and placed into an electric field.

Reviewer 1 sets forth an interesting question. We think that there seems to be a direct relationship between the cell migration level towards the anode and the intensity of the conditioning behavior. However, in order to analyze this phenomenon in a more

detailed way, we want to train cells through successive induction processes and then analyze the resulting learning curves. This way, we will be able to quantify the correlation between the induction process and the conditioned response. We plan to measure the intensity of the conditioned response by calculating the modules of displacement of cell trajectories. A study about learning in amoebas is considered for a next future work that we are planning.

(4) The description of the experiments in the segment between lines 306 and 351 (second and third scenario) is not completely clear. Line 308: “these cells were placed once more in Chalkley’s medium without any stimulus for 3 minutes”. Were these cells physically extracted and replated again? Was the medium exchanged? Or was it just simply that the electric field was turned off for 3 min and then turned on with the inverse polarity? If they were extracted and replated, then the inversion of polarity would not be meaningful. (In this respect, the caption to Fig. 5B is confusing: if cells were extracted and replated, they cannot possibly “remember” the direction of the electric field, and hence, the switching of the direction of the field would not be meaningful.) If cells stayed in place, why was the polarity not inversed without the 3 min break and why the trajectories of cells reversing their direction of migration are not shown?

(4.1) The description of the experiments in the segment between lines 306 and 351 (second and third scenario) is not completely clear.

We have re-written this part of the manuscript. See next detailed comments:

(4.2) Line 308: “these cells were placed once more in Chalkley’s medium without any stimulus for 3 minutes”. Were these cells physically extracted and replated again? Was the medium exchanged? Or was it just simply that the electric field was turned off for 3 min and then turned on with the inverse polarity?

Yes, the cells were physically extracted and replated.

When an *Amoeba proteus* is placed in an electric field for long periods of time (>40 minutes), the probability of dying or, at least, detaching from the substrate and adopting a spherical shape increases sharply. The 3-minute “resting time” we gave to the amoebas was our way to minimize cell damage.

First we changed the medium in the setup; next the cells were again placed in the clean experimental chamber and then exposed to galvanotaxis with inverted polarity with fresh medium for 30 minutes as in the second scenario.

(4.3) If they were extracted and replated, then the inversion of polarity would not be meaningful. (In this respect, the caption to Fig. 5B is confusing: if cells were extracted and replated, they cannot possibly “remember” the direction of the electric field, and hence, the switching of the direction of the field would not be meaningful.)

We think that the amoebas seem to associate the anode with the peptide in the induction process. After the induction process, both stimuli remained linked in some cells for a relatively long period of time, and consequently, the systemic movement of amoebae

responded to the presence of an electric field migrating towards the anode in experiments carried out in absence of peptide.

By inverting the electric field, we demonstrate that the amoebae are neither directed to a specific point in the space nor they associate a specific point in the space to the protein. Instead, they seem to associate the presence of the protein to the positive pole.

(4.5) If cells stayed in place, why was the polarity not inversed without the 3 min break

The cells were physically extracted and replated for 3 minutes to minimize cell damage or even cell dying. We changed the medium in the setup, after that the cells were again placed in the clean experimental chamber, and finally exposed to galvanotaxis with inverted polarity with fresh medium for 30 minutes like in the second scenario.

Considering the intrinsic characteristics of this specific process (to extract and replate for 3 minutes to minimize cell damage), it is not possible to film the cells inverting the direction of migration.

However, the reversal of the direction of the amoebas could be filmed in an experiment with cells that acquired the systemic conditioning behavior after a mere induction process (Fig. 5).

(4.6) and why the trajectories of cells reversing their direction of migration are not shown?

As previously explained, due to the fact that we needed to extract the cells from the experimental chamber and leave them resting to minimize the damage related to long expositions to the electric field, it was impossible to film the cells inverting their direction of migration. However, we have performed this process in an experiment designed specifically for that purpose and depicted in Fig. 5.

(5) Altogether, the experimental results, even when taken at face value, allow the following alternative interpretation. The exposure to nFMLP makes a sub-population of cells change the character of their migration in an electric field, making them migrate towards the anode rather than the cathode. The control in Fig. 5A does not necessarily rule it out, because in this control, cells were exposed to a single concentration of nFMLP of 0.2 nM, whereas it is not clear to what concentrations of nFMLP cells were exposed in the gradient experiment (induction). This change in the character of migration then persists for ~ 1hr, gradually fading away. While being an interesting phenomenon, it does not fit the narrative of Pavlovian associative conditioning.

In all our experiments, we used the same concentration of nFMLP ($2 \times 10^{-6} \text{M}$) in the positive pole well of the second block of electrophoresis. A sentence about this issue has been added in Methods section. We corrected the spelling misprint in concentration value appeared in the manuscript (0.2 nM; see "Peptide gradient" in Methods section).

We agree with the description made by the reviewer 1: a sub-population of cells change the characteristic of their migration within an electric field when they are exposed simultaneously to nFMLP. This change in the character of migration persists for around 1 hour, gradually fading away thereafter.

In our opinion, amoebas seem to associate the anode with the peptide in the induction process. After the conditioning, both stimuli remained linked for a relatively long period of time, and consequently, the systemic movement of amoebae responded to the presence of an electric field by migrating towards the anode instead of the expected migration to the cathode.

For the first time, we have observed a systemic cell behavior that can be modified by two external, simple and independent stimuli, when they are simultaneously applied. This new cellular behavior persists for around 1 hour and there seems to be conditioned by only two simultaneous stimuli.

Pavlov studied four fundamental kind of persistent behavior provoked by two stimuli. In our cellular version of Pavlov's experiments, we have studied one of them, the classical type of Pavlovian conditioning called "simultaneous conditioning", in which the conditioned and unconditioned stimulus are presented at the same time. In continuation of this work, other variants and characteristics of cellular associative conditioning and learning processes are under analysis.

I would like to mention that our experiments on the conditioning in amoebae were based on previous studies that we published earlier. In fact, this work represents the experimental verification of the numeric predictions (computational modellings) that we performed in the past in studies dealing with complex metabolic networks.

So, in 2013 we numerically analyzed complex enzymatic processes under systemic conditions through Statistical Mechanics tools, and advanced Computational and Artificial Intelligence techniques (see References section: De la Fuente et al., 2013, <https://doi.org/10.1371/journal.pone.0058284>). In this study, we were able to verify numerically that enzymatic activities organized in modular metabolic networks are governed by Hopfield-like attractor dynamics, similar to what happens in neural networks. A key attribute of the analyzed metabolic Hopfield-like dynamics is the presence of Pavlovian associative memory. This quantitative study showed that the associative memory in unicellular organisms is possible. Such memory would be a manifestation of emergent properties underlying the complex dynamics of the systemic cellular metabolic networks.

Notice that Hopfield dynamics have been addressed in neural network studies for years, and it is generally named "associative memory" in neuroscience (De la Fuente et al., 2013).

[Redacted]

(6) Some secondary remarks and recaps. The drawings and the description of the experimental setup are unclear. The number labeling is ambiguous and dimensions are not shown in the drawing. There is no characterization of the gradient of nFMLP and of the levels of nFMLP that different cells are exposed to during the "induction". The consistency and repeatability of the gradient between different experiments is not addressed. This issue is particularly relevant, because a stable gradient stretching over 20 mm would be practically impossible to achieve. The

procedures of the extraction and transfer of cells and of their placement in different media are not described.

(6.1) The drawings and the description of the experimental setup are unclear. The number labeling is ambiguous and dimensions are not shown in the drawing.

Figure 1 has been improved and the setup is now described in greater detail in the Methods section. In addition, we have added a set of 9 videos as supplementary material. The labeling has been corrected and the dimensions have been added to Figure 1.

(6.2) There is no characterization of the gradient of nFMLP and of the levels of nFMLP that different cells are exposed to during the “induction”. The consistency and repeatability of the gradient between different experiments is not addressed. This issue is particularly relevant, because a stable gradient stretching over 20 mm would be practically impossible to achieve.

We have confirmed the establishment of the peptide gradient by the direct measurement of fluorescein-tagged peptide concentration with a plate reader (new Fig. 2). The concentration of peptide in the middle part of the glass chamber (where the amoebae are placed) increases immediately following the laminar flow establishment (within 2 minutes the concentration rises from zero to approximately 0.2 μM) and this concentration increases further (to 0.6 μM) for at least 30 minutes.

(6.3) The procedures of the extraction and transfer of cells and of their placement in different media are not described.

I would like to apologize again for the insufficient description of the experimental chamber and the experiment itself across the manuscript. Following the reviewer’s indications, we have included a detailed explanation of this issue in the Methods section. In addition, we have added a set of 9 videos (see supplementary material) describing the experimental setup and the mounting procedure in detail.

Response to Reviewer # 2:

Technical Comments:

From here on, the reviewer’s concerns are colored in blue, while our responses are colored in black.

1. It is not clear from the manuscript how many times the experiments were repeated. For example, in Lines 260-275, 180 cells were imaged, but was this experiment repeated using cells from another cell culture? I am also somewhat concerned about the numbers of cells that were analyzed in Figs. 3C and 3D (i.e., is 25-50 cells sufficient for this type of analysis?). Furthermore, representing each cell as an experimental replicate (i.e., N) is not appropriate as there may be differences between cells from different cultures and this biological variation must be accounted for.

(1.1) It is not clear from the manuscript how many times the experiments were repeated. For example, in Lines 260-275, 180 cells were imaged, but was this experiment repeated using cells from another cell culture?

During the whole experiment, several stock cultures of *Amoeba proteus* had to be made and maintained. All the stock cultures, however, came from the same original culture sent to us by Carolina Biological Supply Company, Burlington, NC. Item # 131306.

In the induction process, as well as scenario 1, the experiments were repeated 32 times. scenarios 2 and 3 were performed 27 and 9 times respectively.

The induction process was usually performed using around 7 cells per experiment, sometimes as few as 3 or 4 and other times as many as 10 or 11, but most of the time the average was 6-8 cells. The number of cells analyzed in the scenarios depended on how many cells appeared to be conditioned in the first step, so that the number of cells per experiment is lower each time, for instance, in scenario 1, the number of cells was usually between 2 and 4. Finally, in scenarios 2 and 3 the experiments were performed using fewer amounts of cells per experiment, usually 1-3.

In order to clarify this issue, these paragraphs have been added in the manuscript (see "Cell preparation" in Methods section).

(1.2) I am also somewhat concerned about the numbers of cells that were analyzed in Figs. 3C and 3D (i.e., is 25-50 cells sufficient for this type of analysis?).

The significance of the results can be calculated by the p-values, which depend on the number of cells employed in each experiment.

For example, the p-values comparing galvanotaxis without induction with the second (50 cells) and the third scenarios (25 cells) were 10^{-11} and 10^{-10} , respectively. A p-value of these orders (10^{-11}) indicates that the probability of obtaining our results by chance is one in a hundred thousand millions. Therefore, the very high significance obtained in scenarios 2 and 3 reinforces that the number of cells was sufficient.

Moreover, valid results with a smaller number of cells have been reported in scientific literature, for instance: [Rosner, B. Fundamentals of Biostatistics Brooks/Cole, 2015].

(1.3) representing each cell as an experimental replicate (i.e., N) is not appropriate as there may be differences between cells from different cultures and this biological variation must be accounted for.

It is true that depending on culture conditions, the cell behavior may display some variations. However, in order to correct the possible environmental differences between cultures, *Amoeba* cells were starved for 24 hours in Chalkley's Simplified Medium (the same medium that was used in the experiments). Once starved, only the amoebae that were strongly attached to the substrate, actively moving through it and showing a little amount of thin pseudopodia were used in the experiments.

This sentence has been added in the manuscript (Method section).

As it was commented above, the induction process was usually performed using around 7 cells per experiment, sometimes as few as 3 or 4 and other times as many as 10 or 11, but most of the time the average was 6-8 cells. The number of cells analyzed in the scenarios depended on how many cells appeared to be conditioned in the first step, so that the number of cells per experiment is lower each time; for instance, in scenario 1, the number of cells was usually between 2 and 4. Finally, in scenarios 2 and 3 the

experiments were performed using the fewer amounts of cells per experiment, usually 1-3.

2. Are these observations unique to *Amoebae proteus*? To my knowledge, other unicellular amoeba do not exhibit this behavior. Given the potential significance of their conclusions, I strongly feel that the author's need to repeat these experiments using another unicellular model. This is not an unreasonable request given the potential significance of the findings and the fact that the authors have already established the experimental setup.

We agree with the reviewer's suggestion. We have reproduced the process with *Metamoeba leningradensis* (a different genus than *Amoeba proteus*) and the results are similar to those obtained with *Amoeba proteus*.

3. Is it possible to repeat these experiments using human cells?

The answer is yes.

First we plan to perform these experiments with mouse cells in a different setup to know more about the cellular learning processes.

Learning is an intrinsic characteristic of any adaptive process both in physiological and in pathological conditions. For instance, human cells display permanent adaptive physiologic phenomena during embryogenesis, organogenesis and tissue repair, all of them crucial steps for human life preservation and evolution. One of the most relevant examples of cellular adaptation is the epithelial to mesenchymal transition (EMT) process. EMT involves a change in the cellular phenotype, shifting from epithelial to mesenchymal, mediated by transcription factors and other proteins like Snail, E-cadherin, N-cadherin, beta-catenin, Slug, Twist, SETD2, and others. Although subjected to permanent debate, the intimate mechanism of this learning/adaptation remains largely unknown.

Carcinogenesis is another critical example in which cellular adaptations via learning may occur, and this phenomenon has very important implications in medicine. A malignant tumor is a community of individuals (cells) interacting one each other by complex cell-to-cell signals. These communities of cells behave following rules defined by Ecology. Intratumor heterogeneity is a constant phenomenon in the temporal evolution of most cancers. The reason for which some regions of a given tumor acquire divergent properties may be an example of adaptation via learning but, again, their intimate mechanisms remain unknown. The acquisition of resistance to some antineoplastic drugs is another example of this learning that is responsible of most therapeutic fails in clinical practice today. A third example of adaptation is the development of metastases in many human cancers, a leading cause of mortality in this disease. For unknown reasons, a specific group of cells develop new properties that allow them to invade blood vessels and travel within the bloodstream until they colonize distant organs in the body. EMT processes play also a role here. Resistance implies adaptation and adaptation implies learning. [Redacted]

We understand that knowing more about the learning process occurring in human cancer cells is of paramount importance in Oncology with potential benefit for humans. For this reason, once demonstrated that learning is an intrinsic process in amoebae, we

are starting the set-up of the experiment to analyze it in normal mouse cells first and in human cancer cells later.

4. In each of the scenarios, why are some of the cells not conditioned? The authors currently provide no insight into this observation.

Yes, in scenario 1 we detected some cells which did not show any apparent sign of conditioning although being appropriately conditioned. We think that, for unknown reasons, cells display a range of differences in their membrane receptors, electric potential and physiological/metabolic functioning. There are no two identical unicellular organisms, and these cells, that the reviewer refers, probably were weakly conditioned due to their intrinsic physiological peculiarities. In the case of scenarios 2 and 3, probably some cells have lost progressively their respective conditionings along the time as a consequence of the aforementioned physiological differences. We have considered this possibility and plan to analyze it in detail in the next future. Especially, we plan to train cells with successive induction processes and then analyze the resulting learning curves. We plan to measure the intensity of the conditioned response by calculating the modules of the displacement of cell trajectories. This way, we will be able to see the relationship between the induction process and the conditioned response more clearly.

5. The authors should include data showing that a gradient of peptide was established during their induction scenario. Perhaps they could label the peptide with a fluorescent molecule such as FITC?

We have confirmed the establishment of the peptide gradient by the direct measurement of fluorescein-tagged peptide concentration with a plate reader (new Fig. 2). The concentration of peptide in the middle part of the glass experimental chamber (when the amoebae are placed) increases immediately following the laminar flow establishment (within 2 minutes the concentration rises from zero to approximately 0.2 μM) and this concentration increases further with the time (to 0.6 μM) for at least 30 minutes. To address this point we have added a new graphic (Fig. 2) and an explanatory text in the Results and Methods sections.

6. Line 382: The authors need to clarify what they mean by “valid cells”. All cells in an experiment are valid (including outliers) and should be included in the analysis. They should not be omitted simply because they show something different than the average cell.

Following the reviewer’s indications, all the cells (including outliers) have been considered valid in the experiments and, consequently, all of them have been included in the respective analyses.

7. Some details about the organism (i.e., *Amoebae proteus*) need to be provided in the Introduction to set the stage for why this organism was used in the study (i.e., why was this organism the best choice for this study?).

Given the robustness in behavior, easy handling in the laboratory and well documented sensitivity to the electric field and diverse substances, we have chosen *Amoeba proteus*

as the experimental study species in our work. The aforementioned sentence has been included in the Introduction section.

8. Line 158: “that enabled to create” should be replaced with “enabled the creation of”

The suggested change has been performed.

9. Lines 266-268: Is it absolutely necessary to express the percentages with 1 decimal point? I don't know if this adds anything meaningful.

In order to give a solution to the reviewer's concern, we have removed the decimal point from every percentage in the text and figures.

10. For clarity, the data in Figure 2 should be presented as follows:

- A. Trace – without stimulus**
- B. Plot of displacement angle for A**
- C. Trace – galvanotaxis**
- D. Plot of displacement angle for C**
- E. Trace – chemotaxis**
- F. Plot of displacement angle for E**

Following the reviewer's suggestion, the proposed changes have been incorporated.

11. Line 288: “located” should be replaced with “deposited”

The suggestion has been incorporated.

12. Line 431: The first letter in “Petri dish” is always capitalized. Please ensure consistency in the text of the manuscript.

Following this indication, we have capitalized the first letter in "Petri dish" through all the manuscript.

13. Line 476-478: The statement here should be moved to the Introduction and expanded upon (see Comment 7).

We have tried to move this statement to the Introduction in different ways and several times. However, each time we did it we realized that the meaning of the sentence appears isolated, disconnected, and out of context. For this reason, we have rewritten the sentence to be more precise and maintaining it in its current place.

The former sentence in the 3rd paragraph of the Discussion was:

In our experiments this conditioned motility pattern prevailed for durations ranging from 20 to 95 minutes. It should be mentioned that the cellular cycle of *Amoeba proteus*, with variations depending on the environment, is usually about 24 hours long under controlled culture conditions¹⁰

The new sentence in the 3rd paragraph of the Discussion reads as follows:

In our experiments, this conditioned motility pattern prevailed for durations ranging from 20 to 95 minutes. This period of time is very long in comparative terms if we take into consideration that the cellular cycle of *Amoeba proteus* lasts, with small variations depending on the environment, only about 24 hours under controlled culture conditions¹⁰.

Response to Reviewer # 3:

Technical Comments:

From here on, the reviewer's concerns are colored in blue, while our responses are colored in black.

1. The use of an electric field as the conditioned stimulus seems highly artificial and of little relevance to the natural behavior of amoeba.

Almost all cellular plasma membranes have an electrical potential. Typical values of membrane potential, normally given in millivolts, range from -40 mV to -80 mV. This potential can vary depending on the cell type, for example muscle cells range between -50 and 60 mV. Changes in the electric membrane potential enable communication with other cells and have multiple physiological functions as, for example in protists predators like amoebas, the detection of prey. In fact, practically all protists and metazoan cells are capable of producing and detecting electrical potentials in the range of physiological values. For instance, these electrical signals are essential and fundamental in neurons for the functioning of the nervous system. Typical values of the resting electric potential range in neurons from -70 to -80 millivolts.

We hope that reviewer 3 will agree with us in that the election of an electric field as the conditioned stimulus is appropriate. Other types of stimuli will be considered in future works that we are planning.

On the other hand, in Pavlov's classic experiments, the conditioned stimulus was the bell's signal, what seems not to correspond to any typical wild signal in Nature. This fact does not detract, however, from his extraordinary scientific finding.

2. Common components of classical conditioning studies include analyses of random or unpaired CS and US presentations, and sensitivity of the magnitude of the conditioning to the timing of the CS and US (i.e., forward and backward pairings). The present study is incomplete without including analyses of these key parametric features of classical conditioning.

The main objective of our study is to demonstrate that *Amoeba proteus* have the capacity to change their systemic behavior when exposed simultaneously to two simple external stimuli. These modifications in the cellular behavior persist during 45 minutes on average, what is a notable temporal space considering that the cellular cycle of *Amoeba proteus* is usually about 24 hours long under controlled culture conditions.

Following the suggestions of reviewers 1 and 2, we have replicated the whole process with a different species, the *Metamoeba leningradensis*, and we have found similar results to those observed in *Amoeba proteus*. In total, we have analyzed in this work the migration trajectories of almost one thousand cells. Also, we have quantitatively studied

the peptide gradient in the setup, we have performed new controls and we have significantly improved the description of the experimental process.

As a consequence, the manuscript has been significantly expanded to reach 9 graphics and 26 panels in total. We are aware that the manuscript represents only the first step in a new area of the knowledge on the systemic behaviors induced in cells. Many more will come.

As described in our manuscript, we have studied a classical type of Pavlovian conditioning called “simultaneous conditioning”, in which both stimuli are presented at the same time. In continuation of this work, other variants and characteristics of cellular associative conditioning and learning processes are under analysis by our research group.

[Redacted]

We hope the reviewer will agree with us that, only a minimally comprehensive study of learning based in the associative conditioning in unicellular organisms, requires at least a specific and detailed study in the form of a new manuscript.

3. Replications of the experimental groups should be provided.

We have replicated the whole process with *Metamoeba leningradensis* and we have added a new control in the manuscript.

4. There is no mention of blind procedures, which are essential in these studies.

We agree that blind procedures are often essential. During this investigation the researchers involved in the quantitative analysis of the cellular trajectories were never aware of what scenario each trajectory belonged to. Only when all the trajectories were correctly quantified and processed, the researchers in charge of recording the amoeba's movements informed the rest of team of which trajectories belonged to each scenario.

5. No mechanistic analyses are presented. Consequently, the results will contribute little to the understanding of memory mechanisms. The development of another simple system with which to analyze memory mechanisms is unlikely to be of great value to the field.

We agree with the reviewer 3 assertions. In fact, we have just finished a depth research in which we have analyzed the movement trajectories of enucleated and non-enucleated *Amoeba proteus* using advanced statistical mechanic techniques.

This work is right now under reviewers in a journal belonging to the Nature group and you can find the manuscript here

<https://drive.google.com/open?id=1d7EXkHMHDkk23CPFL734XOsmwmcBL4Bf>

We believe that this new work complements our Pavlovian study and answers the third reviewer's query.

The main result of this new work is that we demonstrate the existence of a dynamic memory implicated in the correct cellular migration of amoebae. As it is well known, Pavlovian associative memory links two short-term memories (working memories). In this new manuscript, we describe for the first time the memory implicated in cellular migration regulation. More specifically, we have found a dynamic memory of 41.5 minutes of average in all the analyzed cells and cytoplasts, which corresponded to non-trivial correlations lasting approximately 1,245 move-steps. It is worth to consider that this "long-term" memory (mimicking short-term memory in neuronal systems) coincides with the temporal duration of the associative memory observed in our manuscript about the Pavlovian behavior in amoebae in which the conditioned motility patterns prevailed for 45 minutes on average.

This work also remarks the value of the convergence between Quantitative Sciences and Life Sciences, one of the main challenges in Contemporary Scientific Thought. In fact, our study has been carried out by a senior multidisciplinary group composed by specialists in cell biology and physical-mathematical sciences, as for instance, Prof. José I. López with a vast trajectory in Life Sciences research, culminated with three recent articles in *Cell*, and Prof. Terry Sejnowski (<https://es.coursera.org/instructor/terry>), pioneer of research in neural networks, co-promotor of the computational neuroscience, and member of the team who developed the Brain Activity Map Project.

Moreover, in 2013 we also numerically analyzed different enzymatic processes under systemic conditions through Statistical Mechanics tools, and advanced Computational and Artificial Intelligence techniques (see References section: De la Fuente et al., 2013, <https://doi.org/10.1371/journal.pone.0058284>). In this study, we were able to verify numerically that enzymatic activities organized in modular metabolic networks are governed by Hopfield-like attractor dynamics, similar to what happens in neural networks. A key attribute of the analyzed metabolic Hopfield-like dynamics is the presence of Pavlovian associative memory. Hopfield dynamics have been addressed in neural network studies for years, and it is generally named "associative memory" in neuroscience (De la Fuente et al., 2013).

[Redacted]

6. There is excessive nonstandard terminology to describe learning phenomena and procedures (e.g., "non-conditioned stimulus," "loss of persistence," "double stimulus").

Following the reviewer's advice, we have changed the terms "non-conditioned stimulus", "loss of persistence" and "double stimulus" by "unconditioned stimulus", "loss of conditioning" and "induction process", respectively.

7. There is nonstandard terminology to describe statistical tests (e.g., line 216, the behaviors "were absolutely different").

We have changed “were absolutely different” by “were significantly different”.

Reviewers' comments:

Reviewer #2 (Remarks to the Author):

Introduction

I would still like to see some additional information in the Introduction about the organisms used in the study (e.g., some discussion about the life cycle, habitat, key discoveries made using the organism, etc). For example, it's still unclear why *Amoeba proteus* was selected as the experimental system. There are many unicellular organisms that are easily handled in the lab and display robust directional responses to different environmental cues. What makes *Amoeba proteus* especially useful for this study?

Results, Section 4

So, based on their results, would the authors conclude that the chemotactic gradient (i.e., food) has a stronger influence on cell migration than the electric field? If so, then this is somewhat confusing given the findings presented in Fig. 3, which suggest that the attraction towards the electric field is stronger (i.e., essentially all cells migrate towards the cathode in a strong directional manner, whereas the migration towards the peptide is not as directed). Can the authors please provide some clarification on this in the text of the manuscript?

At the end of this section, why is Fig. 7 referenced before Figs. 5 and 6? Figures should be referenced in the order of their appearance in the text. Can the authors please correct this?

Results, Section 5

With regards to the sentence "In the first scenario, 85 new amoebae that had previously migrated...": If these amoeba "previously migrated towards the anode-peptide", then how can they be considered "new"? The authors tend to mis-use the word "new" in the manuscript and should ensure that only those cells that were never exposed to any of the conditions employed in the study are referred to as "new".

When stating the percentages associated with the conditioned behavior, the authors should include the error associated with these measurements (e.g., 58% +/-...).

Discussion

The authors still do not provide any insight as to why some cells were not conditioned. This should be commented on in the Discussion. For example, a brief summary based on their response in the rebuttal document should be included in the Discussion section.

Minor grammatical corrections

Title: Should be "Evidence" not "Evidences"

Revise

"Given the robustness in behavior, easy handling in the laboratory and well documented sensitivity to the electric field and diverse substances, we have chosen *Amoeba proteus* as the experimental study species in our work."

To the following

"Given the robustness in their behavior, easy handling in the laboratory, and well documented sensitivity to electric fields and chemoattractants, we have chosen *Amoeba proteus* as the experimental study species in our work."

Revise

"Interestingly, we have performed a preliminary study in another unicellular species, *Metamoeba leningradensis*, which also exhibited this systemic conditioned behavior."

To the following

"We have also performed a preliminary study in another unicellular species, *Metamoeba leningradensis*, which also exhibited this systemic conditioned behavior."

Revise

"In Fig. 7, a galvanotactic control of the cells that responded to the cathode during the induction process is showed. All the amoebae migrated towards the cathode, confirming that these cells were unconditioned."

To the following

"In Fig. 7, a galvanotactic control of the cells that responded to the cathode during the induction process is shown. All the amoebae migrated towards the cathode, confirming that these cells were unconditioned."

Reviewer #3 (Remarks to the Author):

The manuscript has not been revised satisfactorily.

1. Common control procedures (e.g., Rescorla, 1967) and parametric analyses for classical conditioning studies have not been performed as requested. Without these, the paper remains incomplete. And importantly without the controls, the paper cannot claim to have demonstrated Pavlovian associative conditioning. Either the claim needs to be removed from the paper or the proper controls need to be performed.
2. The issue of replications has not been clarified. By replications this reviewer did not mean using another experimental system, but rather, and similar to the concern of Reviewer #2, the number of experimental replications. In response to Reviewer #2, the authors indicate that for the induction process about 7 cells were used "per experiment." If a total of 180 cells were used as reported in the Results, then the N should be about 26. But the reader never sees this number presented and it appears that the authors are inappropriately using the total number of cells as the N. If so, there are no replications. Similar issues apply to the cells analyzed in each of the "scenarios." Well-established classical conditioning procedures using groups of subjects (e.g., 150 *Drosophila*) trained simultaneously treat each group experiment as a separate N (e.g., Tully and Quinn, 1985). A similar approach is used for group training of *C. elegans* (A. J. Yu et al., *Current Protocols in Neuroscience*, 86, e57).
3. Simply presenting P values for the Wilcoxon rank-sum tests is insufficient. The authors need to provide the complete descriptive statistics for each test. E.g., as described in: http://evc-cit.info/psych018/Reporting_Statistics.pdf: A Mann-Whitney test indicated that self-rated attractiveness was greater for women who were not using oral contraceptives (Mdn = 5) than for women who were using oral contraceptives (Mdn = 4), $U = 67.5$, $p = .034$, $r = .38$.
4. Blind procedures are not discussed in the revised manuscript.
5. Absence of mechanistic analyses. The authors indicate such analyses will be forthcoming in future work, but their absence in the present manuscript is a weakness and consequently the results of the present paper contribute little to the understanding of memory mechanisms.

Response to Reviewer # 2:

Technical Comments:

From here on, the reviewer's concerns are colored in blue, while our responses are colored in black.

(1) Introduction

I would still like to see some additional information in the Introduction about the organisms used in the study (e.g., some discussion about the life cycle, habitat, key discoveries made using the organism, etc.). For example, it's still unclear why *Amoeba proteus* was selected as the experimental system. There are many unicellular organisms that are easily handled in the lab and display robust directional responses to different environmental cues. What makes *Amoeba proteus* especially useful for this study?

Following the indications of the reviewer #2, we have included in the "Introduction Section" the additional information:

"Amoebae represent an immensely diverse family of eukaryotic cells that can be found in nearly all habitats and constitute the major part of all eukaryote lineages⁶. Concretely, Amoeba proteus is a large free-living predatory amoeba with a notable capacity to detect and respond to chemical and physical cues allowing it to locate and consume near prey organisms such as bacteria and other protists.

These cells are able to migrate on flat surfaces and in three-dimensional substrate by a process known as amoeboid movement, which consists in pseudopodia extensions, cytoplasmic streaming, and flowing into these extensions changing permanently the cellular shape⁷.

Amoeboid locomotion represents one of the most widespread forms of cell motility and constitutes the typical way of locomotion in broad range of adherent and suspended eukaryotic cell types⁷. In mammalian cells, amoeboid locomotion is vital for multiple physiological processes as the development of the embryo⁸, the action of the immune system⁹ and the repair of wounds¹⁰. Likewise, it is also responsible for the spread of malignant tumors¹¹.

The large free-living amoeba, Amoeba proteus, has served as a classic unicellular organism in many investigations for more than one hundred years¹²⁻¹³, mainly as a cellular model to study cell motility, membrane and cytoskeleton function, and the role of the nucleus¹⁴⁻¹⁶. However, despite the many investigations carried out so far, numerous biological aspects of this organism still remain poorly studied. On the other hand, diverse experimental studies have shown that Amoeba proteus exhibit robust galvanotaxis¹⁷, a directed movement in response to an electric field; in fact, it has been described that practically 100% of the amoebae migrate towards the cathode for long periods of time under a strong direct-current electric field in a range between 300 mV/mm and 600 mV/mm. Likewise, amoebae are known to display chemotactic behaviors; in particular, the peptide nFMLP, typically secreted by bacteria, is able to provoke a strong chemotactic response in many different types of cells. The presence of this peptide in the environment may indicate to the amoeba that food organisms might

be nearby¹⁸. Given the large number of investigations carried out on this organism, the robustness in their behavior, easy handling in the laboratory, the relatively fast rate of migration (cells move at approximately 300 $\mu\text{m}/\text{min}$ ¹⁹) and the well documented sensitivity to electric fields and chemoattractants, we have chosen *Amoeba proteus* as the experimental study species in our work.”

(2) Results, Section 4

2.1 So, based on their results, would the authors conclude that the chemotactic gradient (i.e., food) has a stronger influence on cell migration than the electric field? If so, then this is somewhat confusing given the findings presented in Fig. 3, which suggest that the attraction towards the electric field is stronger (i.e., essentially all cells migrate towards the cathode in a strong directional manner, whereas the migration towards the peptide is not as directed). Can the authors please provide some clarification on this in the text of the manuscript?

2.2 At the end of this section, why is Fig. 7 referenced before Figs. 5 and 6? Figures should be referenced in the order of their appearance in the text. Can the authors please correct this?

2.1

Reviewer # 2 asks if we may conclude that the chemotactic gradient is stronger than the electric field. Using the results obtained in our experiments, we cannot draw that conclusion. Concretely, Figure 3 illustrates that 86% of the *Amoeba proteus* showed a positive chemotactic response while 100% of the cells responded adequately to the electric field (migrated towards the cathode); also, the galvanotactic response appeared to be slightly more intense, as can be seen in Fig.3. However, though it may seem at first sight that the galvanotactic response is stronger in cells influenced by one stimulus, a new “non-linear” response of cell sub-population arises when the amoebae are subjected to both stimuli simultaneously. In this new population behavior about half the amoebae cells migrated towards the anode (where was the nFMLP peptide), and approximately the other half of the cellular population migrated to the cathode. On the other hand, 39% of the *Metamoeba leningradensis* moved towards the anode-peptide, in the induction process (Figure S1). Therefore, in a strict sense, it is not possible to predict the outcome of an induction process (Figure 4a) based on the responses observed when the stimuli were separated (Figure 3b and 3c).

In order to address this issue raised by the reviewer, we have included the following text in page 26:

*“When the cells were subjected to both stimuli at the same time (induction process), a new population response arose. This population behavior, in *Amoeba proteus*, showed that about half the amoebae cells migrated towards the anode (where the nFMLP peptide was placed), and approximately the other half of the cellular population migrated to the cathode (Figure 4). On the other hand, only 39% of the *Metamoeba leningradensis* moved towards the anode-peptide, in the induction process (Figure S1). Accordingly, the cellular migration response under two simultaneous stimuli is notoriously different from that observed when the stimuli were separated (Figure 3b, 3c and 8a), and therefore it cannot be concluded that the chemotactic gradient has a stronger influence on cell migration than the electric field.”*

2.2

The requested correction has been made.

(3) Results, Section 5

3.1 With regards to the sentence “In the first scenario, 85 new amoebae that had previously migrated...”: If these amoeba "previously migrated towards the anode-peptide", then how can they be considered "new"? The authors tend to mis-use the word “new” in the manuscript and should ensure that only those cells that were never exposed to any of the conditions employed in the study are referred to as “new”.

3.2 When stating the percentages associated with the conditioned behavior, the authors should include the error associated with these measurements (e.g., 58% +/-...).

3.1

We agree with the reviewer’s suggestion and all these “news” have been removed.

3.2

We would like to clarify this issue. In order to express our results, instead of averaging the percentage obtained in each experimental replication and expressing its error (as the referee mentions), we decided on reporting the total percentage of cells, considering all the experiments together. This decision was made because the number of cells per replication could not be fixed, since it depended on the previous experiment (for example, the number of cells used in the first conditioning test of the first scenario depended on the number of the induced cells). Therefore, instead of averaging percentages of experiments with different sample sizes (which could be hard to interpret, and would need a weighting correction)¹, we considered that in this case reporting the total percentage would be clearer for the reader. Besides, given the close relationship of the percentage with the mean of a distribution, it is well known that, when the size of the sample is big and the quantities that are averaged are normally distributed, the means are normally distributed, and for smaller sizes it follows the Student’s t-distribution, but this cannot be used in our case, because the different means that we are using do not need to come from the same probabilistic distribution.

1-Knapp, T. R., Allen, W., Berra, Y., & Edison, T. (2009). Percentages: The most useful statistics ever invented. Retrieved May, 10, 2016.

(4) Discussion

The authors still do not provide any insight as to why some cells were not conditioned. This should be commented on in the Discussion. For example, a brief summary based on their response in the rebuttal document should be included in the Discussion section.

On this issue the following text has been added in the discussion section:

“We have also observed that, after the induction process, a small subset of the amoebae was not conditioned. Cells display a range of differences in their membrane receptors, electric potential, physiological/metabolic functioning, and hence, there are no two

identical unicellular organisms. In our experiments, some cells probably were unconditioned or weakly conditioned due to their intrinsic physiological peculiarities, and in addition, some kind of cellular damage caused by the experimental process may have occurred.”

(5) Minor grammatical corrections

Title: Should be “Evidence” not “Evidences”

Revise

“Given the robustness in behavior, easy handling in the laboratory and well documented sensitivity to the electric field and diverse substances, we have chosen *Amoeba proteus* as the experimental study species in our work.”

To the following

“Given the robustness in their behavior, easy handling in the laboratory, and well documented sensitivity to electric fields and chemoattractants, we have chosen *Amoeba proteus* as the experimental study species in our work.”

Revise

“Interestingly, we have performed a preliminary study in another unicellular species, *Metamoeba leningradensis*, which also exhibited this systemic conditioned behavior.”

To the following

“We have also performed a preliminary study in another unicellular species, *Metamoeba leningradensis*, which also exhibited this systemic conditioned behavior.”

Revise

“In Fig. 7, a galvanotactic control of the cells that responded to the cathode during the induction process is showed. All the amoebae migrated towards the cathode, confirming that these cells were unconditioned.”

To the following

“In Fig. 7, a galvanotactic control of the cells that responded to the cathode during the induction process is shown. All the amoebae migrated towards the cathode, confirming that these cells were unconditioned.”

All the “minor grammatical corrections” have been added.

Response to Reviewer # 3:

Technical Comments:

From here on, the reviewer’s concerns are colored in blue, while our responses are colored in black.

(1) Common control procedures (e.g., Rescorla, 1967) and parametric analyses for classical conditioning studies have not been performed as requested. Without these, the paper remains incomplete. And importantly without the controls, the paper cannot claim

to have demonstrated Pavlovian associative conditioning. Either the claim needs to be removed from the paper or the proper controls need to be performed.

- We have removed from the manuscript any statement about our results corresponds to the classical Pavlovian associative conditioning.

- In addition, to make this issue completely clear, we have added the following paragraph in the discussion section:

“However, in a strict sense, we cannot conclude that our findings represent the classical Pavlovian conditioning since complete controls and parametric analyses for classical conditioning studies have not been performed yet²³.”

This reference (23) is the one provided by the reviewer # 3

- Finally, we changed the title of the manuscript:

Before:

“Evidences of Pavlovian associative conditioning in *Amoeba proteus*”

Now:

“Evidence of conditioned behavior in *Amoeba proteus*”

(2) The issue of replications has not been clarified. By replications this reviewer did not mean using another experimental system, but rather, and similar to the concern of Reviewer #2, the number of experimental replications. In response to Reviewer #2, the authors indicate that for the induction process about 7 cells were used “per experiment.” If a total of 180 cells were used as reported in the Results, then the N should be about 26. But the reader never sees this number presented and it appears that the authors are inappropriately using the total number of cells as the N. If so, there are no replications. Similar issues apply to the cells analyzed in each of the “scenarios.” Well-established classical conditioning procedures using groups of subjects (e.g., 150 *Drosophila*) trained simultaneously treat each group experiment as a separate N (e.g., Tully and Quinn, 1985). A similar approach is used for group training of *C. elegans* (A. J. Yu et al., *Current Protocols in Neuroscience*, 86, e57).

In order to clarify the reviewer’s concern, we have included the information regarding the number of experimental replicates, and the amount of cells used per experiment both in the text of manuscript and in all figures. In all experimental sections of our work this information has been added. The new notation is the following: “N” for total number of cells used in the experiment; “Er” for number of experimental replicates; “nr” for number of cells used in each replicate.

(3) Simply presenting P values for the Wilcoxon rank-sum tests is insufficient. The authors need to provide the complete descriptive statistics for each test. E.g., as described in: http://evc-cit.info/psych018/Reporting_Statistics.pdf: A Mann-Whitney test indicated that self-rated attractiveness was greater for women who were not using oral contraceptives (Mdn = 5) than for women who were using oral contraceptives (Mdn = 4), $U = 67.5$, $p = .034$, $r = .38$.

We have updated all our statistics according to the reference indicated by the reviewer for the case of Wilcoxon rank-sum test (i.e., p-value and Z-score).

Therefore, now all our statistics appear as, for instance, $p = 10^{-26}$; $Z = -10.639$, where p denotes the p-value and Z indicates the Z-Statistic corresponding to the Wilcoxon rank-sum test.

Besides, we have updated the corresponding text in Methods section explaining this issue, and included an additional reference (reference 26) where the calculation of this statistic and the appropriateness of the test are extensively deepened.

26. Gibbons, J. D., & Chakraborti, S. *Nonparametric statistical inference*. (Springer, 2011).

(4) Blind procedures are not discussed in the revised manuscript.

The text required by the reviewer that was added to the previous rebuttal has been added to the manuscript (Methods Section. p. 28):

“Researchers involved in the quantitative analysis of the cellular trajectories were never aware of what scenario each trajectory belonged to. Only when all the trajectories were quantified and processed, the researchers in charge of recording the amoeba’s movements informed the rest of team of which trajectories belonged to each experiment or control.”

(5). Absence of mechanistic analyses. The authors indicate such analyses will be forthcoming in future work, but their absence in the present manuscript is a weakness and consequently the results of the present paper contribute little to the understanding of memory mechanisms.

We agree with the fundamental aim raised by the reviewer #3. As we explained in the previous rebuttal, an extensive mechanistic study has already been carried out, and another work is already being done. In addition, as we have reflected in the manuscript, the experiments we show here were inspired by mechanistic analysis that we published in 2013 dealing with complex metabolic networks (reference 24). Sincerely, we think that to understand the basic mechanisms of the associative memory in unicellular organisms, in an approximate way, will require many more studies.

REVIEWERS' COMMENTS:

Reviewer #3 (Remarks to the Author):

The manuscript has been revised satisfactorily.